# The relationship between PM$_{2.5}$ and anticyclonic wave activity during summer over the United States

Ye Wang[1, 2], Natalie Mahowald[2], Peter Hess[3], Wenxiu Sun[3, 4], and Gang Chen[5]

[1]College of General Aviation and Flight, Nanjing University of Aeronautics and Astronautics, Nanjing, Jiangsu, China
[2]Department of Earth and Atmospheric Science, Cornell University, Ithaca, NY, USA
[3]Department of Biological and Environmental Engineering, Cornell University, Ithaca, NY, USA
[4]Now at: BloomSky Inc., Burlingame, CA, USA
[5]Department of Atmospheric and Oceanic Sciences, University of California, Los Angeles, Los Angeles, CA, USA

**Correspondence:** Ye Wang (wangye08@nuaa.edu.cn)

**Abstract.** To better understand the role of atmospheric dynamics in modulating surface concentrations of fine particulate matter (PM$_{2.5}$), we relate the anticyclonic wave activity (AWA) metric and PM$_{2.5}$ data from the Interagency Monitoring of Protected Visual Environment (IMPROVE) data for the period of 1988-2014 over the US. The observational results are compared with hindcast simulations over the past two decades using the National Center for Atmospheric Research-Community Earth System Model (NCAR CESM). We find that PM$_{2.5}$ is positively correlated (up to R=0.65) with AWA changes close to the observing sites using regression analysis. The composite AWA for high aerosol days (all daily PM$_{2.5}$ above the $90^{th}$ percentile) shows a similarly strong correlation between PM$_{2.5}$ and AWA. The most prominent correlation occurs in the Midwestern US. Furthermore, the higher quantiles of PM$_{2.5}$ levels are more sensitive to the changes in AWA. For example, we find the averaged sensitivity of the $90^{th}$ percentile PM$_{2.5}$ to changes in AWA is approximately three times as strong as the sensitivity of $10^{th}$ percentile PM$_{2.5}$ at one site (Arendtsville, Pennsylvania; 39.92 °N, 77.31°W). The higher values of the $90^{th}$ percentile compared to the $50^{th}$ percentile in quantile regression slopes are most prominent over the northeastern US. In addition, future changes in US PM$_{2.5}$ based only on changes in climate are estimated to increase PM$_{2.5}$ concentrations due to increased AWA in summer over areas where PM$_{2.5}$ variations are dominated by meteorological changes, especially over the western US. Changes between current and future climates in AWA can explain up to 75% of PM$_{2.5}$ variability using a linear regression model. Our analysis indicates that higher PM$_{2.5}$ concentrations occur when a positive AWA anomaly is prominent, which could be critical for understanding how pollutants respond to changing atmospheric circulation, as well as developing robust pollution projections.

## 1 Introduction

Particulate matter less than 2.5 $\mu$ m in diameter (PM$_{2.5}$) poses a considerable air quality concern due to its impacts on human health (Liu et al., 2020). PM$_{2.5}$ has been linked to increased possibility of mortality (Krewski et al., 2009). Continuing exposure to PM$_{2.5}$ can exacerbate existing cardiovascular and respiratory problems, and cause lung damage (Bernard et al., 2001). It can also alter the body's defense system against foreign materials, and even lead to premature death (Kappos et al., 2004).

Furthermore, PM$_{2.5}$ could contribute to the degradation of visibility (Hand et al., 2011; Ashley et al., 2015) and the alteration of the hydrological cycle through changing rainfall formation mechanisms in clouds (Rosenfeld et al., 2008). Once deposited onto snow PM$_{2.5}$ can cause an increase in melting of snow and ice (Painter et al., 2007) or modify land or ocean biogeochemistry (Mahowald et al., 2011). It also influences Earth's energy balance directly by scattering solar incoming radiation back into space (Charlson et al., 1992) or indirectly by altering cloud albedo and lifetime (Albrecht, 1989; Arimoto, 2001). Aerosols can also alter the large-scale atmospheric circulation and lead to historical changes to wintertime midlatitude winter extremes over northern Eurasia (Wang et al., 2020). Thus, understanding and predicting potential changes in PM$_{2.5}$ is crucial to both human health concerns and current environmental issues.

Climate change and meteorological factors may have substantial impacts on surface concentrations of PM$_{2.5}$. A number of studies have demonstrated the influences of weather and climate change on PM$_{2.5}$ variability (Brasseur et al., 2006; Liao et al., 2006; Dawson et al., 2007; Jacob and Winner, 2009; Thishan Dharshana et al., 2010; Tai et al., 2012; Liu et al., 2017; Chen et al., 2018; Wang et al., 2019; Wang and Zhang, 2020), showing connections between specific meteorological conditions and PM$_{2.5}$ concentration responses. Daily variation in temperature, rainfall, moisture, and circulation can explain up to 50% of the variability in PM$_{2.5}$ (Tai et al., 2010). Meanwhile, Aw and Kleeman (2003) calculated a reduction in PM$_{2.5}$ concentrations caused by rising temperature over southern California in a modeling study. Furthermore, Tai et al. (2012) identified cyclone passage with the associated cold front as a mechanism for temperature being strongly correlated with inter-annual variability of PM$_{2.5}$ in the Midwestern US. Similarly, increased atmospheric stagnation may also have the potential to aggravate PM$_{2.5}$ air quality in the future climate (Liao et al., 2006; Leibensperger et al., 2008). Mickley et al. (2004) also proposed an increased severity of summertime PM episodes due to a warmer future climate in the Northeastern and Midwestern US. In addition, a significantly negative correlation caused by the longer lifespan of gas-phase-produced sulfate was found between aerosol sulfate and cloud cover compared to aqueous-phase-produced sulfate (Koch et al., 2003). These studies suggest that changes in meteorology linked to climate modification may cause variations of PM$_{2.5}$ levels and exposure risks. However, questions remain regarding whether and how PM$_{2.5}$ concentrations directly related to atmospheric general circulation. Understanding the relationship between meteorology and PM$_{2.5}$ levels will be critical to the understanding of pollutant response to a changing circulation due to climate change, as well as the development of robust pollution projections.

Many studies have focused on relating wave activity with extreme weather and climate events. Weather extremes are strongly influenced by the natural variability of the atmosphere at synoptic, intra-seasonal, and inter-annual time scales. For example, persistent high pressure blocking systems can cause extreme cold winter temperatures in Europe (Woolings et al., 2008) or summer heat waves (Coumou et al., 2015). Mid-latitude weather extremes can be influenced by the major modes of climate variability such as Arctic Oscillation (Michel and Rivière, 2011) or El Niño–Southern Oscillation (ENSO) (Ryoo et al., 2013). Furthermore, blocking anticyclones can decay by releasing accumulated wave activity as a stationary Rossby wave train (Takaya and Nakamura, 2001). Meanwhile, blocking highs are connected with high-amplitude, quasi-stationary anticyclonic anomalies that result in protracted and unusual weather events (Nakamura et al., 1997). In addition, temperature extremes are more likely to connect with wave events through the large-amplitude troughs and ridges (Pfahl and Wernli, 2012; Chen et al., 2015; Martineau et al., 2017). Shen and Mickley (2017) identify the association between warm tropical Atlantic sea surface

temperatures and enhanced subsidence, reduced precipitation and increased temperatures through stationary wave propagation in the Eastern US. However, no previous studies have investigated how wave activity can lead to changes in $PM_{2.5}$ concentrations, although studies have shown a robust correlation between surface ozone concentration and a measure of wave activity over the US using a linear regression model (Sun et al., 2019).

Here we propose the use of AWA, the anticyclonic part of local finite-amplitude wave activity (LWA) with quantile regressions, to be an effective method for diagnosing the tendency and sensitivities in the transport of $PM_{2.5}$ concentrations, using a similar approach to a previous study focused on ozone (Sun et al., 2019). LWA assesses longitude-by-longitude anomalies of the finite-amplitude Rossby wave activity deviating from the circle of equivalent latitude using the meridional displacement of potential vorticity (PV) (Huang and Nakamura, 2016). As an example, the prominent Northern Hemisphere blocking episode that occured in late October 2012 was well explained using LWA (Huang and Nakamura, 2016). Furthermore, recent modeling studies have illuminated the value of quantile regression in examining the influences of global changes on local air quality. For example, Porter et al. (2015) employ quantile regression to diagnose the meteorological sensitivities of higher ozone and $PM_{2.5}$ levels by using observed and reanalysis meteorological data in the US over the past decade.

While meteorology sets the stage for the occurrence of dangerous pollutant levels in the present and future climates, the connection between the meteorological events and extreme pollution events is still not sufficiently understood (Dawson et al., 2014). In the present study, we apply a univariate linear regression model to analyze daily $PM_{2.5}$-AWA relationship from observations and simulations in the US, using a similar methodology as that used to examine LWA and ozone relationships (Sun et al., 2019). We also use quantile regression to calculate $PM_{2.5}$ sensitivities to AWA across quantiles from 10% to 90%. In addition, the coefficients of the slope for the linear regression model between $PM_{2.5}$ and AWA in the current climate are evaluated to see how much they can be used to predict future $PM_{2.5}$ changes. Such an exploration of $PM_{2.5}$ from atmospheric motion can provide new insights into understanding the mechanism of $PM_{2.5}$ changes, as well as the sensitivity of higher $PM_{2.5}$ concentrations to climate change.

## 2 Methodology

In order to quantify the relationship between AWA and surface $PM_{2.5}$ concentrations during the summer months, five different combinations of datasets, one observational and four model output, are used in this study. Each of the five cases use surface $PM_{2.5}$ concentrations and geopotential heights (for calculating a modified AWA, as explained below). Each modeling dataset uses 20 years for the analysis, while the observations are analyzed for the years between 1988 and 2014. Both $PM_{2.5}$ and AWA time series are detrended and deseasonalized by eliminating the 21-day smoothed seasonal cycle from the original data. We use the resulting stationary residuals to focus on the synoptic-scale variability, minimizing aliasing from regular seasonal variations or long-term tendencies. This study is a followup study of Sun et al. (2019), using the same approach used there to analyze ozone, and here we focus on analyzing $PM_{2.5}$. The details of the methods are discussed in more detail in that paper, and summarized here.

## 2.1 Observational Data

We use $PM_{2.5}$ measurements from the Interagency Monitoring of Protected Visual Environment (IMPROVE) monitoring sites, which are located in National Parks and Class I Wilderness areas across the US (http://vista.cira.colostate.edu/improve/). $PM_{2.5}$ concentrations are reported every 3 days as mass per air volume at local temperature and pressure. Hand et al. (2011; 2012) have described the details regarding IMPROVE site locations, sampling, analysis approach, and detailed information of network operations. All stations with at least 1000 valid $PM_{2.5}$ values between 1988 and 2014 are collected in this study, totaling 150 stations for summer (June, July, August, JJA hereafter). For the $PM_{2.5}$ observations (called Obs case), the full names, states, latitudes and longitudes, as well as short names, are listed in Table S1 and locations are shown in Figure 1. We chose three representative stations in different parts of the country to investigate the relation between AWA and $PM_{2.5}$ in detail. The IMPROVE station names are AREN1 (Arendtsville, Pennsylvania; 39.92 °N, 77.31°W; in the Northeast), SIPS1 (Sipsey Wilderness, Alabama; 34.34°N, 87.34°W; in the Southeast) and LAVO1 (Lassen Volcanic NP, California; 40.54°N, 121.58°W; in the West), which are shown with the red dots in Figure 1. They match with the site PSU106 (40.72°N, 77.93°W, in the Northeast), SND152 (34.29°N, 85.97°W, in the Southeast) and LAV410 (40.54°N, 121.58°W, in the West) respectively, which has differing impacts of meteorological persistence on the distribution and extremes of ozone in Sun et al. (2017) to allow comparison between the ozone and $PM_{2.5}$ response to AWA. Long range transport from Asia and meteorology are dominant drivers of pollutants at LAVO1, where anthropogenic influence is at a minimum as a clean air site in California (VanCuren and Gustin, 2015).

In order to compare these in situ observations to the large scale meteorology, the 500 hPa geopotential height (m) from the European Renalysis Interim version is used (ERA-Interim) for the time period of January 1991 to December 2010 (Dee et al., 2011). The ERA-Interim is a global reanalysis of recorded meteorological data over the past 3.5 decades and was undertaken by the European Centre for Medium-Range Weather Forecasts. This gridded dataset is created at approximately 0.7° spatial resolution with 37 vertical levels.

## 2.2 Model output

We compare the IMPROVE data with output from Community Earth System Model (CESM) as simulated for the Chemistry-Climate Model Iniative (CCMI) (Eyring et al., 2013. As a state-of-the-art Earth system modeling framework coordinated by the National Center for Atmospheric Research (NCAR), the CESM employed here is configured to fully couple the Community Atmosphere Model version 4 (CAM4) (Neale et al., 2010), the Community Land model version 4.0 (CLM4.0) (Oleson, 2010), the Parallel Ocean Program version 2 (POP2) (Smith et al., 2010) and the Los Alamos sea ice model (CICE version 4) (Hunke and Lipscomb, 2008). All simulations are performed under current land cover conditions. The top of the simulated atmosphere reaches to 40 km with a horizontal resolution of 2.5° longitude by 1.9° latitude. The model has been widely used for modeling the Earth's past, present and future climate states (Neale et al., 2010; Hurrell et al., 2013).

To simulate the atmospheric chemistry, we include the CAM4-Chem module, which has been widely studied on its representation of atmospheric chemistry in the atmosphere (Aghedo et al., 2011; Lamarque et al., 2011a, b, 2012; Lamarque and

Solomon, 2010). CAM4-Chem employs the bulk aerosol model (BAM) with externally mixed aerosols considering black carbon, organic carbon, sulfate, sea spray and desert dust, and which simulates coarse mode aerosols in 4 bins for the latter two.

Details of this implementation are discussed in Lamarque et al. (2012). CAM4 uses the Zhang–McFarlane deep convection scheme (Zhang and McFarlaneb, 1995), Hack shallow cumulus scheme (Hack et al., 2006), Holtslag and Boville's (1993) planetary boundary layer process, and the parameterization of cloud microphysics and macrophysics by Rasch and Kristjánsson (1998) and Zhang et al. (2003). Additionally, the convective momentum transport is utilized to parameterize deep convection (Richter and Rasch, 2008). These two major revisions caused improvements in such aspects as the Madden-Julian oscillation

and ENSO (Neale et al., 2008). The improved trend and magnitude of surface $PM_{2.5}$ using this free-running model has been evaluated elsewhere (Tilmes et al., 2016).

The chemical emissions and forcing details for each of the model simulations are listed in Table 1. The simulation using specified dynamics (REFC1SD) for current levels of $PM_{2.5}$ from 1991 to 2010 is driven by analyzed meteorological data from Modern-Era Retrospective Analysis for Research and Applications (MERRA) (see Tilmes et al., 2016). This simulation

follows the conventions of the CCMI (Eyring et al., 2013). For the AWA analysis for this case, we use the 500 hPa geopotential height from MERRA, which should be very similar to that from ERA-Interim, since they use largely the same observations. To compare the relationship between AWA and $PM_{2.5}$ concentrations in online simulations, three simulations forced by trace gas projections and an interactively coupled ocean are employed. The GCM2000 and GCM2100 simulations are 25-year runs branched from the CCMI reference simulations in the year 2000 and the year 2100, respectively. Simulations over the first

five years are discarded as spin-up, and results from the latter 20 years are discussed here (2006-2025 for GCM2000 and 2106-2125 for GCM2100). Note that while the $CO_2$ concentrations in the GCM2000 and GCM2100 simulations are kept at the year 2000 and 2100 level respectively, the concentrations of all other greenhouse gases including methane remain constant at the year 2000. In particular, the emissions and prescribed chemical species for longer-lived substances follow the protocol defined by CCMI hindcast simulations for the year 2000 (Eyring et al., 2013), which are repeated for all the simulated model

145  years. Another future run (REFC2) is forced by future climate combined with future emissions following the REFC2 CCMI modeling protocol. In this run greenhouse gas forcing and emissions following the RCP6 scenario. The relationship between ozone and AWA has been examined in the GCM2000, GCM2100 and REFC2 simulations in Sun et al. (2019). Characteristics of the REFC1SD simulation are given in Phalitnonkiat et al. (2018). Note that our REFC2 set-up covers volcanic eruptions in the past, but possible volcanic eruptions in the future are not included (Eyring et al., 2013).

## 2.3  AWA calculation

To calculate AWA, we adopt the procedures in Chen et al. (2015) and Huang and Nakamura (2016). A dynamical quantity, q (here we use $Z_{500}$, geopotential height at 500 hPa), approximately decreases with latitude in the Northern Hemisphere. For a given value of q = Q, we introduce an equivalent latitude $\phi_e(Q)$ as

$$\phi_e(Q) = \arcsin[1 - \frac{S(Q)}{2\pi a^2}] \tag{1}$$

Here, S(Q) is the area bounded by the Q contour towards the North Pole and a denotes Earth's radius. Defining an eddy term as $\hat{q} \equiv$ q-Q and separating the southward and northward displacements in the Q contour, we calculate the cyclonic (southern), anticyclonic (northern) and total LWA at the longitude $\lambda$ and latitude $\phi_e$ by

$$A_C(\lambda, \phi_e) = \frac{a}{\cos \phi_e} \int_{\hat{q} \leq 0, \phi \leq \phi_e(Q), \lambda = const} \hat{q} \cos \phi d\phi \tag{2}$$

$$A_A(\lambda, \phi_e) = \frac{a}{\cos \phi_e} \int_{\hat{q} \geq 0, \phi \geq \phi_e(Q), \lambda = const} \hat{q} \cos \phi d\phi \tag{3}$$

$$A_T(\lambda, \phi_e) = A_C - A_A \tag{4}$$

Studies on finite-amplitude wave activity (FAWA) have identified the link between the pattern of atmospheric circulation and large-scale wave dynamics (Nakamura and Solomon, 2011; Methven, 2013; Chen and Plumb, 2014; Lu et al., 2015). LWA adds the longitude dimension to the zonally average quantity FAWA and is calculated from the meridional displacement of quasigeostrophic PV from zonal symmetry (Nakamura and Zhu, 2010). LWA helps differentiate longitudinally isolated events and describe extreme weather events at the local scales (Huang and Nakamura, 2016, Chen et al., 2015). Chen et al. (2015) used local finite-amplitude wave activity based on the 500 hPa geopotential height for characterizing mid-latitude weather events. The total wave activity is composed of the cyclonic wave activity residing to the south of the equivalent latitude and the anticyclonic wave activity to the north (see Fig. 1 in Sun et al, 2019). In this study, we focus on AWA to characterize its connection with changes in PM$_{2.5}$ concentrations. Over the US in summer LWA is dominated by its anticyclonic component (Sun et al., 2019). Sun et al. (2019) also used AWA to characterize ozone variability.

## 2.4 Quantile regression

Quantile regression is used to estimate the slopes for several conditional quantile functions (Koenker and Bassett, 1978). It characterizes the connection between a range of predictor variables and specified percentiles (or quantiles) of the response variable. For example, Porter et al. (2015) analyzed the sensitivities of ozone and PM$_{2.5}$ concentrations for response quantiles ranging from 2 to 98%. The parameters of quantile regression models evaluate the change in a specific quantile of the response variable caused by a one-unit change in the predictor variable. This permits us to measure how some percentiles of the PM$_{2.5}$ may be more influenced by AWA than others, and this is indicated by changes in the regression coefficient. In order to illustrate the sensitivity of the PM$_{2.5}$ concentration at different quantiles, we apply linear quantile regression for percentiles from $10^{th}$ to $90^{th}$ at the AREN1 site. And then we compare the $90^{th}$ percentile quantile regression coefficient with $50^{th}$ percentile quantile regression coefficient at each station.

## 2.5 The univariate linear regression model

To help explore and measure the likely relationship between AWA and PM$_{2.5}$ levels, we use the univariate linear regression model, similar to a previous study focused on ozone (Sun et al., 2019). Here the slope of PM$_{2.5}$ with respect to wave

activity ($S_{i0,j0}(i,j)$) on the daily time scale is used to show the linear association between changes (in time) of the normalized PM$_{2.5}$ at a point $(i_0, j_0)$ and the normalized wave activity at another point. We use the projection of PM$_{2.5}$ onto AWA to reveal how closely the AWA anomaly field resembles the spatial pattern that enhances PM$_{2.5}$ on the daily time scale during the summer. The projection of AWA $(p_{i0,j0})$ at all points in the domain onto $S_{i0,j0}$ is defined according to the following equation:

$$p_{i0,j0} = AWA \cdot S_{i0,j0} = \sum_j \sum_i AWA(i,j) \times S_{i0,j0}(i,j) \tag{5}$$

The similarity between AWA spatial pattern and the PM$_{2.5}$-AWA regression coefficients' spatial structure is estimated by the projection value. The interannual change in PM$_{2.5}$ due to changes in AWA is predicted based on a linear regression model as following equation (see Sun et al., 2019 for more discussion):

$$PM_{2.5} = \beta p + \alpha \tag{6}$$

The change of PM$_{2.5}$ (denoted by $\Delta PM_{2.5}$) in the future due to the change in AWA is calculated using the following equation, that is the climatological difference in future climate and the present climate ($\overline{AWA}_f - \overline{AWA}_p$), project it onto S, and multiply it by the slope $\beta$:

$$\Delta PM_{2.5} = \beta[(\overline{AWA}_f - \overline{AWA}_p) \cdot S] \tag{7}$$

Here, S is calculated from the values for the present climate. The projected value can measure the similarity between the AWA change and the PM$_{2.5}$'s trend with AWA by compressing the information of the AWA field into a single variable. This variable incorporates the non-local effect of AWA on PM$_{2.5}$'s variability.

## 2.6  The composite methodology

We use a composite methodology which is based around the most polluted ($>90^{th}$ percentile) daily PM$_{2.5}$ and the corresponding anomalies at every station. Composite 500 hPa geopotential height and AWA for daily values of PM$_{2.5}$ larger than $90^{th}$ percentile are produced by separately averaging all daily anomaly values of the corresponding 500 hPa geopotential height and AWA. The composite methodology can average out much of the variability. Composite Madden–Julian Oscillation cycles of precipitation and ozone for each phase are examined by averaging together all daily anomaly values for the given quantity separately (Sun et al., 2014). In addition, the meteorological conditions conductive to a high ozone event are investigated by compositing about the first day of each high ozone event in the Northeastern region (Sun et al., 2017).

## 3  Results and discussion

The monthly mean PM$_{2.5}$ surface concentrations with standard deviation for different scenarios at three representative sites are shown in Figure 2. PM$_{2.5}$ concentrations are largest during summer at the three sites. The climatological average for

PM$_{2.5}$ is greater in the Eastern than in the Western sites. A statistically significant correlation (r>0.80, p<0.01; Figure 2) for current PM$_{2.5}$ is found between observations and simulations of monthly mean climatological averages (REFC1SD and GCM2000) at three representative sites. The highest correlation coefficients between model and observations (0.93) are seen at SIPS1 perhaps due to the large seasonal variation in PM$_{2.5}$ concentrations (Figure 2b). The future PM$_{2.5}$ concentrations are increased in GCM2100 under current emissions compared with current climate PM$_{2.5}$ simulations. There is a strong decease in climatological mean for future PM$_{2.5}$ at AREN1 under future emissions and meteorology (REFC2), while the climatological average for future PM$_{2.5}$ has no significant change under future emissions at SIPS1 and LAVO1. Such differences in the monthly mean averages for PM$_{2.5}$ suggests that emission changes are more important than climate changes at AREN1, but it is not clear which is more important at SIPS1 or LAVO1.

Focusing on the spatial distribution, the highest PM$_{2.5}$ concentrations over the 20-year average in summer occur in the south-central US (Figure 3a; green lines; GCM2000 case). The 20-year averaged AWA on summer days exhibits a maximum over the Southwestern US (Figure 3a; shaded). The difference between two current climate simulations (REFC1SD minus GCM2000) for summertime AWA is shown in Figure 3b. The reduced AWA in the reanalysis forced simulation is found across most of the US, with the largest reduction in the Southwestern US as previously shown (Sun et al., 2019). In contrast, the AWA in summer is higher over the Northeastern US in the forced simulations. The corresponding changes in summertime PM$_{2.5}$ concentration caused by a combination of different emissions and possibly changes in AWA is similar to changes in AWA, although the reduction is largely over South-central US. The difference between two future scenarios (GCM2100; REFC2) and current climate scenario (GCM2000) has a similar pattern (illustrated in Figure 3c and d), which shows a large increase in AWA in the Southwestern US, but there is a difference in the amplitude of these changes (contrast Figure 3c vs. 3d) (Sun et al., 2019). There is an increase in PM$_{2.5}$ concentration for the future scenario with current emissions (GCM2100), while there is a decrease in PM$_{2.5}$ concentrations when future emissions are used (REFC2), showing the importance of future potential decreases in emissions.

## 3.1 Relationship between PM$_{2.5}$ concentrations and AWA at specific stations

For the three observation sites highlighted here (AREN1, SIPS1 and LAVO1), the PM$_{2.5}$ concentrations are positively cor-related with AWA in the areas close to the sites where presumably at least some of emissions of PM$_{2.5}$ are located (Figure 4). The relationships between daily PM$_{2.5}$ concentrations and AWA using CESM simulations presented here offer a test of the consistency between observational and model relationships in characterizing the response of PM$_{2.5}$ to AWA. The highest regression coefficient occurs in the observational (Obs) and the reanalysis driven simulated cases (REFC1SD), as opposed to the case coupled metereology (GCM2000) (Figure 4a, b, c (top row) and Figure 4d, e, and f (middle row) in contrast to Figure 4g, h and I (bottom row)). The highest spatial regression coefficients for site AREN1 and SIPSl are located southward of the sites, while they are located to the northwest at LAVO1. Overall the model simulates similar spatial patterns to the observations for the case of the reanalysis driven simulations (REFC1SD), but do less well for the coupled model simulations (GCM2000). Of course, model predictions are not perfect, and include uncertain emissions and boundary conditions, as well as errors in model physical and chemical processes, which may be driving these inconsistencies between modeled and observed relation-

ships. In addition, Sun et al. (2019) shows considerable variation in the AWA pattern between different ensemble simulations, suggesting even on the timescale of 20 years there is considerable internal variability between model ensemble members.

To obtain a more in-depth understanding of the physical mechanisms behind the relationship between $PM_{2.5}$ concentration and AWA, we consider the composite AWA for high $PM_{2.5}$ days at AREN1, SIPSl and LAVO1 (Figure 5). The pattern for the composite AWA corresponding to daily $PM_{2.5}$ above the $90^{th}$ quantile is most similar to regression coefficients between daily AWA and $PM_{2.5}$ by comparing different quantiles (Figure S1). The composite AWA calculated by averaging together all AWA corresponding to daily $PM_{2.5}$ above the $90^{th}$ quantile shows a similar strong connection between $PM_{2.5}$ and AWA as that seen for the average (Figure 4 vs. 5). The composite AWA are extremely pronounced over the areas where $PM_{2.5}$ data originates. The geopotential composites for the highest pollution days generally showed spatial distributions which are similar to the regression coefficient distributions (Figure 5 vs. 4) at the three representative sites. The areas with the largest values of the composite AWA are located southward of the AREN1 and SIPSl sites. But at LAV01 the maximum is located to the northwest for the observational and reanalysis driven cases (Obs and REFC1SD), and eastward for the coupled model case (GCM2000). Overall, the composite AWA for $PM_{2.5}$ also shows that the daily $PM_{2.5}$ above its $90^{th}$ quantile connects strongly with AWA during summer. Note that there is a spatial displacement between maximum of geopotential height and the maximum of AWA, since wave activity fundamentally measures the waviness of atmospheric general circulation, rather than the magnitude.

## 3.2 Relationship of AWA and $PM_{2.5}$ regionally

Next we consider how the local relationship between $PM_{2.5}$ and AWA changes in space. To simplify the visualization of the spatial variability in the local relationship, we use the result from the previous section that the maximum regression coefficients between $PM_{2.5}$ and AWA are usually close to the site where $PM_{2.5}$ is measured (Figure 5). The highest composite AWA anywhere in the domain and the highest regression coefficient with AWA are shown at each gridpoint in Figure 6. If we look at the relationship between the $PM_{2.5}$ concentration and the wave activity at each location, it can be seen that $PM_{2.5}$ concentrations are positively correlated with AWA throughout the US, but with varying strengths (Figure 6). A roughly similar spatial distribution is obtained when either the composite AWA for high $PM_{2.5}$ (left-hand side: Figure 6a, c, e, g and i) and for the regression coefficients between the $PM_{2.5}$ and AWA (right-hand side: Figure 6b, d, f, h and j) for the different observational or model combinations, showing the consistency of the approaches. The pattern of the distribution is consistent for observational (Figure 6a and 6b) and simulated (Figure 6c and 6d) data when forced with reanalyses winds, with the largest values in the upper mid-west into the mid-Atlantic states. The climate model simulations tend to have a stronger correlation in the western/mountain regions, than seen in the reanalyses winds (Figure 6c vs. 6e), and there are some hints of this in the observations in Arizona, for example (Figure 6a). The correlations and composites tend to become stronger in the western/mountain regions in the future model simulations (Figure 6g and 6i vs. Figure 6e).

The consistency in the composite AWA under high $PM_{2.5}$ conditions and the correlations argues that either of these metrics can be useful tools to identify $PM_{2.5}$ and AWA relationships. The anticyclonic condition is usually characterized by, low-level divergence, subsiding air, light wind, no rainfall, and high surface pressure. Taken together, the results above demonstrate the positive connection of $PM_{2.5}$ concentrations with anticyclonic conditions everywhere across the US, which is likely accounted

for arid weather and sinking inversions. This is consistent with the $PM_{2.5}$ sensitivity study of Tai et al. (2010). They illustrated strong linkage between high $PM_{2.5}$ concentrations, high 850 hPa geopotential height and stagnation which is characterized by an anticyclonic condition. The significant association of $PM_{2.5}$ with stagnation is also demonstrated by Cheng et al. (2007) in their examination of four Canadian cities. Furthermore, similar results were reported that a greatly strengthening ozone is associated with increased stagnation (Wu et al., 2008; Sun et al., 2014).

The composite AWA is for $PM_{2.5}$ that larger than the $90^{th}$ quantile, while the regression coefficient is for all $PM_{2.5}$. The composite AWA and regression coefficient have similar spatial distributions suggesting the positive connection between daily $PM_{2.5}$ and AWA is mainly produced by high $PM_{2.5}$ concentration above its $90^{th}$ quantile. Overall, the relationship between behavior of AWA and extreme $PM_{2.5}$ concentration is generally consistent with the existing meteorological studies (Woolings et al., 2008; Coumou et al., 2015; Michel and Rivière, 2011; Ryoo et al., 2013).

### 3.3 The sensitivity of quantiles in $PM_{2.5}$ concentrations to AWA

To examine the sensitivity of different levels of $PM_{2.5}$ concentrations to AWA, we fit the linear regression and quantile regression for AWA and daily $PM_{2.5}$ for summers between 1988 to 2014 from IMPROVE monitoring sites for different percentiles ($10^{th}$ to $90^{th}$ percentiles) using an "impact region" of AWA at AREN1 site. Here the averaged AWA over the "impact region" is defined as an elliptic area bounded by the maximum and minimum longitude and latitude of the maximum composite AWA

for $PM_{2.5}$ larger than $90^{th}$ percentile minus the 0.05 contour line (blue elliptic circle in Figure 5a). One can clearly see that the higher percentiles of $PM_{2.5}$ are more sensitive to the change in the averaged AWA over the impact region, e.g., the $90^{th}$ percentile of $PM_{2.5}$ is approximately three times more sensitive to the averaged AWA over the impact region when compared with the $10^{th}$ percentile of $PM_{2.5}$ (Figure 7a). The correlation coefficient of 0.36 between JJA deseasonalized $PM_{2.5}$ and impact region's average AWA implies that the vast majority of all variability is being driven by factors other than AWA at the AREN1

site. It must be noted that this lack of overall correlation implies other drivers of $PM_{2.5}$ variability at sites like this.

In order to examine whether the relationship that high quantile of $PM_{2.5}$ is more sensitive to the AWA than the low quantile of $PM_{2.5}$ applies to the other sites, we calculate the difference of $90^{th}$ percentile quantile regression coefficient (slope) from $50^{th}$ percentile quantile regression coefficient at the 5% significance level across all sites ($90^{th}$ percentile quantile regression coefficient (slope) minus $50^{th}$ percentile quantile regression coefficient, shown in Figure 7b). Out of the 150 sites, 145 sites

show that $90^{th}$ percentile $PM_{2.5}$ increases more than the $50^{th}$ percentile of $PM_{2.5}$ with the enhancement of the AWA. In the Northeast region (north and east of New York state with New York state included), this relationship is the most pronounced. This difference in response between the highest and median $PM_{2.5}$ values indicates the different sensitivities within various percentiles of the $PM_{2.5}$ levels. These results are to some extent consistent with those from Porter et al. (2015), which addressed the greater sensitivities to mean daily temperature at the highest concentration percentiles in predicting summertime $PM_{2.5}$,

but with $PM_{2.5}$ sensitivities to temperature peak entirely in the east due to the regionality of $PM_{2.5}$ speciation.

## 3.4 Projected PM$_{2.5}$ concentrations due to changes in future AWA

The strong association between PM$_{2.5}$ concentrations and AWA in the current climate prompts us to investigate the extent to which we can utilize a linear regression model to predict changes in PM$_{2.5}$ concentrations from AWA change in future climate. Employing daily present-day summertime concentrations of PM$_{2.5}$ and AWA for current climate from the coupled model simulation (GCM2000, 2006-2025) and equation (5)-(6), we derive that how much of PM$_{2.5}$'s interannual variance can be explained by the projection of JJA AWA anomalies onto the daily PM$_{2.5}$-AWA regression coefficients pattern. The coefficient of determination ($R^2$) of the linear regression model using simulated PM$_{2.5}$ and AWA for the present climate varies from 0 to 0.75 depending on gridbox (Figure 8). This means that the projected value (using only AWA changes) captures up to 75% of the interannual variability in PM$_{2.5}$ over Great Plains and West. Wise and Comrie (2005) similarly determined $R^2$ values of 0.1-0.5 for associations of PM with atmospheric variables across sites in the Southwest, and here we see a comparable relationships across the Southwest, although these studies use different methodologies as well as consider different time periods. Because of the high correlation coefficients (75%) this suggests that the regression results reveal the broad population instead of a small number of influential outliers (Cook, 1979). The $R^2$ measures the part of variance of PM$_{2.5}$ that can be explained by the linear regression model (Kutner et al., 2004).

Next we explore how much of the future change in PM$_{2.5}$ concentrations can be predicted just on the basis of changes in AWA. Using PM$_{2.5}$-AWA relationships determined from current coupled model output (GCM2000), future PM$_{2.5}$ changes can be estimated by using the linear relationship fitted with the current data and projected change of AWA in the future (as shown in equation (5)-(7)). Here we assume that the linear relationship between the predictors and PM$_{2.5}$ do not change very much in the future compared to the present to extrapolate the current linear relationship between PM$_{2.5}$ and AWA to the future. Future climate change is simulated to cause an increase in PM$_{2.5}$ concentrations over most of the US if there are no changes in emissions (GCM2100; Figure 9a). Using CAM4-chem fitted slope pattern and regression coefficient, enhanced PM$_{2.5}$ is found over most of the western US and a small area in the Northeast (Figure 9b). The prediction suggests that the significant increase in future PM$_{2.5}$ resulted from AWA changes arises over the western US, which is up to 0.92 $\mu$ g $m^{-3}$. Great Plains, South-central, Midwest and Southwest show a small decrease of PM$_{2.5}$ in the future, where a negative value of the projection value occurs when projecting the positive anomaly onto it. The projection of PM$_{2.5}$ change in most parts of the South is less reliable, because of the low $R^2$ in this region in the interannual variability metrics (Figure 8).

In order to investigate how much change in PM$_{2.5}$ can be caused by the change of AWA, the fraction of predicted JJA PM$_{2.5}$ change in total JJA PM$_{2.5}$ change from simulation is calculated in Figure 10. Over all, the maximum values are found in the western US. We infer that AWA can be generally utilized in PM$_{2.5}$ predictions during summer in the US, especially over the western US. The impact of AWA change alone on summer PM$_{2.5}$ concentrations is likely to be quite significant (above 50%) in the western US, suggesting that AWA is a suitable tool for air quality predictions for most regions where meteorology dominates. These results are somewhat consistent with Tagaris et al. (2007) who showed that the Midwest was modelled to have the larger daily average PM$_{2.5}$ levels in the future, but our signal is in the west. In addition, studies suggest increases in

PM$_{2.5}$ over polluted regions in the future climate caused by intensified stagnation (Liao et al., 2006; Jacob and Winner, 2009),

which is consistent with the anticyclonic condition seen with AWA in this study.

  Future changes in 500 hPa JJA geopotential height anomaly between the future simulation (GCM2100) relative to the current simulation (GCM2000) (shown in Figure S2) can account for the PM$_{2.5}$ variability resulting from changes in AWA. The 500 hPa geopotential height increases everywhere by approximately 25-55 m over the entire US. These strengthened geopotential height can be explained by mid-to-high latitude warming in the future climate. The increased geopotential height at higher

latitudes is consistent with other model projections (Vavrus et al., 2017). The increase in 500 hPa height which shows a distinct anticyclonic pattern centering over the western US and the adjacent ocean is consistent with changes in a suite of atmospheric variables related to changes in PM$_{2.5}$ concentrations (Dawson et al., 2007; Tai et al., 2010; Porter et al., 2015).

  Increased AWA over most parts of US in the future climate are projected to increase PM$_{2.5}$ levels over western regions where meteorology dominates fluctuations in PM$_{2.5}$. This is consistent with some studies reporting an increased PM$_{2.5}$ concentrations

due to more common and extended stagnation periods across northern mid-latitudes in the future climate if anthropogenic emissions remain constant (Mickley et al., 2004; Liao et al., 2006; Leibensperger et al., 2008; Jacob and Winner, 2009). Similarly in simulations, a mean increase of 0.24 $\mu$ g $m^{-3}$ in summer average PM$_{2.5}$ levels with a largest growth of 0.93 $\mu$ g $m^{-3}$ in the Midwest has been shown (Tai et al., 2010). An increase in PM$_{2.5}$ despite globally increasing precipitation is also obtained by using coupled chemistry climate models, revealing a decreased precipitation on a large scale across polluted

regions and seasons (Fang et al., 2011; Kloster et al., 2010). In contrast, Avise et al. (2009) found that changes in meteorology tend to reduce summertime PM$_{2.5}$ concentrations (approximately -1 $\mu$ g $m^{-3}$) in most regions with the maximum reductions over the Southeastern US caused by intensified wet deposition. Furthermore, some models suggest there could be a regional increase in summertime PM$_{2.5}$ over the eastern US due to lower precipitation (Racherla and Adams, 2006), while Tagaris et al. (2007) found that climate change, alone, with no emissions increase or controls affects the US PM$_{2.5}$ concentrations slightly.

The discrepancy between studies mentioned above and the results given here is most likely attributable to differences in model formulation. Although earlier studies predict important changes in PM$_{2.5}$ levels in a warming climate throughout the US, there is no consistency across studies (Jacob and Winner, 2009). The unpredictable sensitivity of PM$_{2.5}$ levels to climate change could be explained by the complication of the reliance of different PM$_{2.5}$ elements on climatic variables, and the uncertainties in regional boundary layer ventilation and precipitation, as well as the diversity of PM$_{2.5}$ components (Racherla and Adams,

2006; Pye et al., 2009).

  It should be noted that the change in surface PM$_{2.5}$ predicted by the future change of AWA using the univariate linear regression models is different from the simulated future change of PM$_{2.5}$ from the constant emissions run, which simulate the most significant change in the eastern US (Figure 9a), consistent with the other model projections (Tai et al., 2012). This discrepancy mostly results from the distribution of projected AWA change (Figure 3). Figure 9b only includes the change

caused by the change of AWA. These studies did not account for the change of PM$_{2.5}$ separately due to stagnation, temperature or other meteorological conditions, which could also act a significant part in the PM$_{2.5}$ changes. In addition, the predicting capability of the linear models is limited, and the model only looks at linear relationships between PM$_{2.5}$ levels and AWA. Also, it only looks at the mean of PM$_{2.5}$ levels.

PM$_{2.5}$ generally consists of multiple different aerosols each with different sources and variability; for example, the most important in the US are sulfate, organic matter, elemental carbon, nitrate, ammonium and desert dust. The different PM$_{2.5}$ components respond to meteorological variables differently. The sulfate fraction of PM$_{2.5}$ is predicted to be higher due to faster SO$_2$ oxidation under a warmer climate while the nitrate and organic fraction lower due to volatility (Dawson et al., 2007; Kleeman, 2008; Tai et al., 2010). Increased temperatures can lead up to higher biogenic emissions of PM$_{2.5}$ precursors including agricultural ammonia, soil NO$_x$, and volatile organic compounds (Pinder et al., 2004; Bertram et al., 2005; Guenther et al., 2006; Riddick et al., 2016). Aqueous-phase sulfate and ammonium nitrate production increase with higher relative humidity (Liao et al., 2006; Dawson et al., 2007). Wildfires are an important source of black and organic carbon and they can increase or decrease depending on the local changes in climate and land use (Park et al., 2007; Spracklen et al., 2009; Kloster et al., 2012). Future exploration of the different components of aerosols and how each responds to climate could provide more information about the effect on each type, but for these simulations, only PM$_{2.5}$ was output and thus is not available for this study.

## 4   Conclusions

We employed a univariate linear regression model to determine the correlation of PM$_{2.5}$ levels and AWA on synoptic-scales over the US. This analysis demonstrates that PM$_{2.5}$ is positively linked to the local anticyclonic finite-amplitude wave activity over the past two decades during JJA, and the high PM$_{2.5}$ concentrations are more sensitive to the AWA than those low ones. The relationship between AWA and PM$_{2.5}$ in model-simulated and observational data agrees in their general pattern and amplitude. These results provide insights into the drivers behind high PM$_{2.5}$ pollution episodes in the observed record, emphasizing the significance of atmospheric circulation to the pollutant accumulation.

We found that AWA is positively correlated with PM$_{2.5}$ at every available station in the summer using regression analysis in this study. The most prominent relationship between PM$_{2.5}$ and AWA occurs in the Midwestern US for current climate, while it moves westward in the future climates. The composite AWA for PM$_{2.5}$ larger than its $90^{th}$ percentile can also demonstrate the positive relationship between PM$_{2.5}$ and AWA. Climate change in the future is likely to cause a response in regional PM$_{2.5}$. The sensitivities of PM$_{2.5}$ levels to changes in AWA are, generally, more robust for higher percentiles through quantile regression, which is most prominent in the northeastern US. It means that changes to AWA are likely to influence the extent of PM$_{2.5}$ extremes more strongly than they influence moderate PM$_{2.5}$ levels. This study presents new perspectives to explore both the observed and simulated PM$_{2.5}$ responses to climate change. Furthermore, the contrast of observed sensitivities to those simulated by CESM could determine essential model biases relating to the prediction of future PM$_{2.5}$, potentially offering perceptions into the fundamental mechanistic reasons behind those biases.

The coefficient of determination of the linear regression model using simulated PM$_{2.5}$ and AWA for the present climate is up to 75% over Great Plains and West, which shows that the daily variation in AWA can project up to 75% of interannual PM$_{2.5}$ variability across the US. These effects suggest that AWA could have significant impacts on the PM$_{2.5}$ levels. Significant

regional variation is found in these results, indicating that while the positive association between $PM_{2.5}$ and AWA are generally consistent, the extent to which AWA influences $PM_{2.5}$ is local.

*Data availability.* $PM_{2.5}$ data are available at http://vista.cira.colostate.edu/improve. ERA-Interim data can be accessed at http://www.ecmwf. int/en/research/climate-reanalysis/era-interim. The CCMI output data can be downloaded from the Centre for Environmental Data Analysis
(http://data.ceda.ac.uk/badc/wcrp-ccmi/data/CCMI-1/output/).

*Author contributions.* YW analysed the observational data and visualized the model output. YW and NM wrote the manuscript. PH and WS provided critical feedback, and helped shape the research and analysis. GC provided the expertise on local finite-amplitude wave activity. All authors reviewed the manuscript.

*Competing interests.* The authors declare that they have no conflict of interest.

*Acknowledgements.* We acknowledge high-performance computing support from Cheyenne (doi:10.5065/D6RX99HX) provided by NCAR's Computational and Information Systems Laboratory, sponsored by the National Science Foundation. We also thank Simone Tilmes.

*Financial support.* This research has been supported by NUAA grants (NS2019037) and the National Science Foundation (NSF AGS-1608775).

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

**Table 1.** The model case names used in this study. Designated model descriptions for the four attribution cases. Online means that the model is calculating the meteorology prognostically online. Case names are based on names described previously (Eyring et al., 2013; Sun et al., 2019).

| Simulation (years) | Model Case Name | $GHG^1$ forcing | $SST^2$ and sea ice | Emissions | Meterological driver |
|---|---|---|---|---|---|
| REFC1SD (1991-2010) | f.e11.TSREFC1SD.f19.f19.ccmi23.001 | $CMIP5^3$ | $HadISST2^4$ | Anthropogenic and biomass burning emission: $MACCity^5$ Biogenic emissions: $MEGAN2^6$ | $MERRA^7$ |
| GCM2000 (2006-2025) | b.e11.TSREFC2.femis2000.y2000.f19.f19.ccmi23.001 | $CO_2 = 369$ ppm | $Online^8$ | Anthropogenic and biomass burning from $AR5^9$ Biogenic emissions: Monthly values from MEGAN2 for 2000 | Online |
| GCM2100 (2106-2125) | b.e11.TSREFC2.femis2000.y2100.f19.f19.ccmi23.001 | $CO_2 = 669$ ppm | $Online^8$ | Same as GCM2000 | $Online^8$ |
| REFC2 (2080-2099) | b.e11.TSREFC2.f19.g16.ccmi23.001.cam.h7 | A1B scenario | Online | A1B scenario | Online |

[1]Greenhouse gas. [2]Sea surface temperature. [3]Coupled Model Intercomparison Project. [4]Hadley Centre Sea Ice and Sea Surface Temperature data set (Titchner and Rayner, 2014). [5]Granier et al. (2011). [6]Guenther et al. (2012). [7]Modern-Era Retrospective Analysis for Research and Applications (Rienecker et al., 2011). [8]Tilmes et al. (2016). [9]Assessment report 5 (Eyring et al., 2013)).

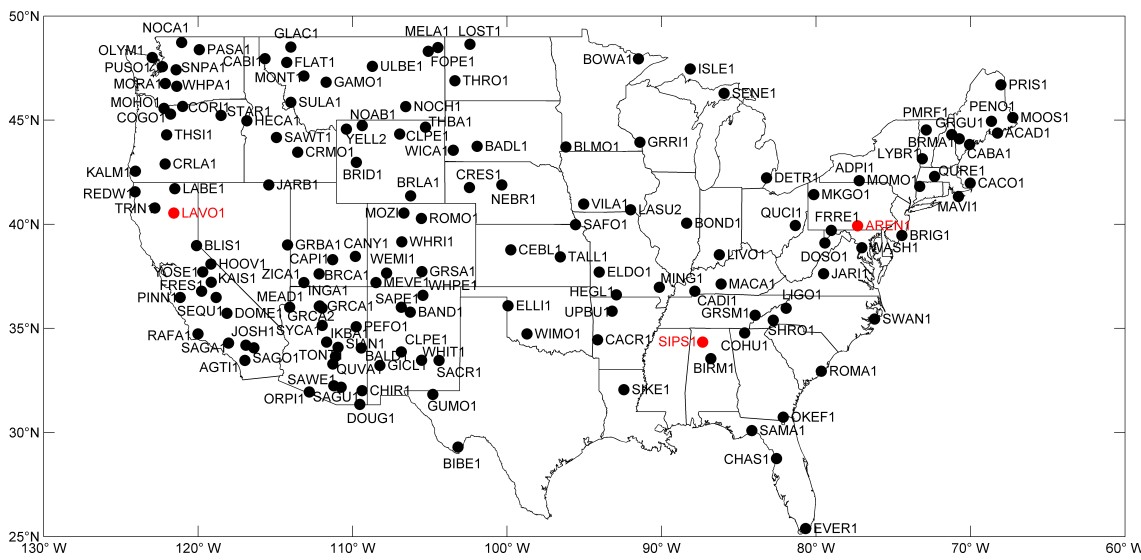

**Figure 1.** Geographical location of IMPROVE sites for PM$_{2.5}$. Red dots are three representative sites.

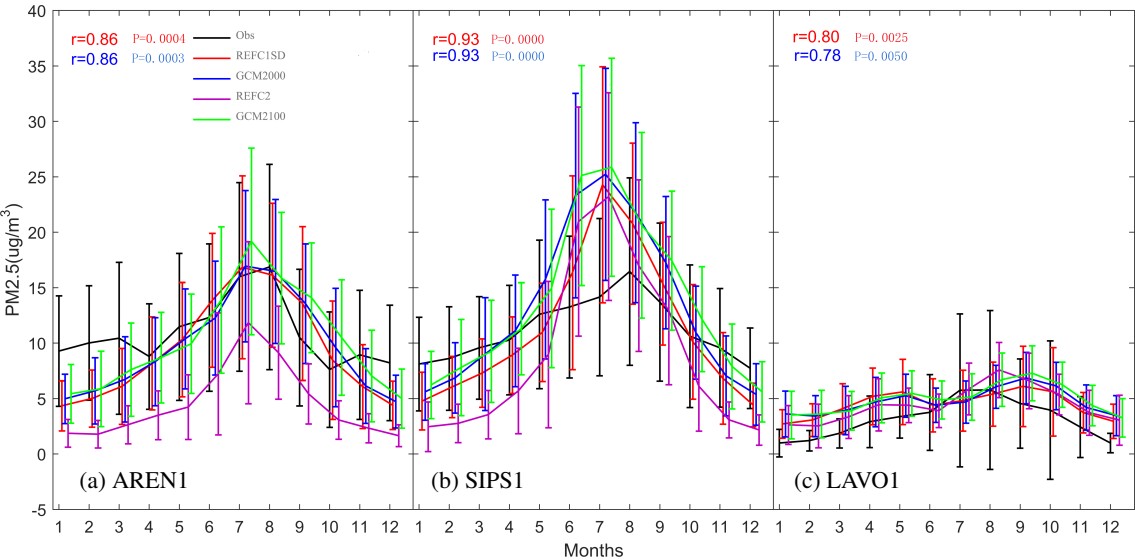

**Figure 2.** Climatological monthly mean average with standard deviation for PM$_{2.5}$ used in this study at three sites (AREN1 (a), SIPS1 (b) and LAVO1(c)) for five scenarios used in this study. Red (r) represents the correlation coefficient between observation (Obs) and simulation for current from the REFC1SD simulation. Blue r represents the correlation coefficient between observation (Obs) and simulation for current from GCM2000 simulation. The p-values are included.

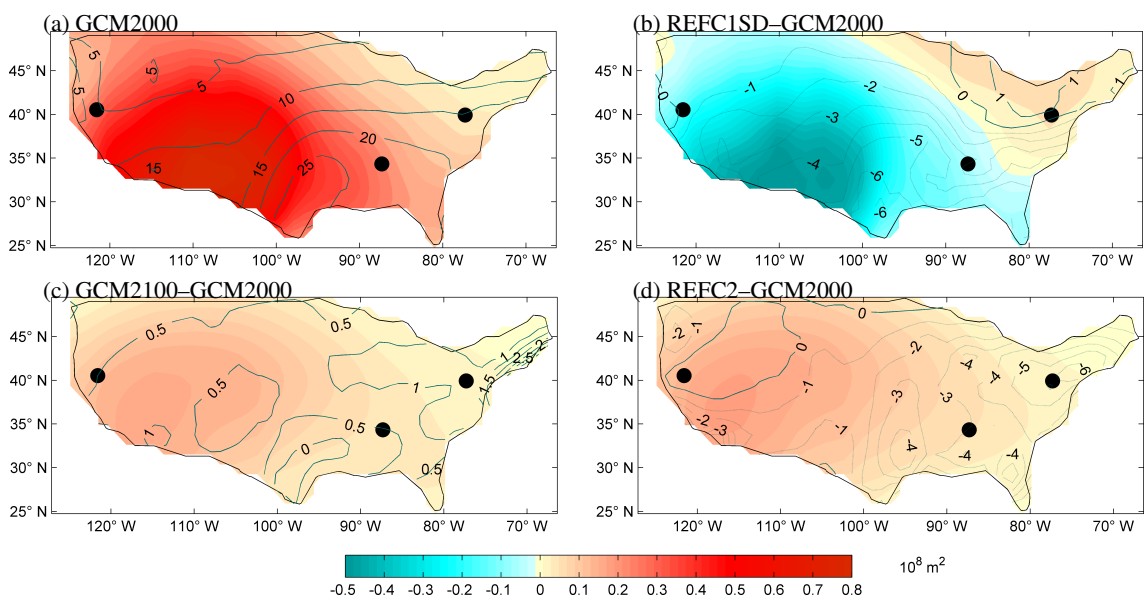

**Figure 3.** Wave activity (AWA: shaded using the legend in $10^8\ m^2$) and PM$_{2.5}$ concentrations (green contour lines in $\mu$ g $m^{-3}$) for (a) the current climate (GCM2000, 2006-2025 summer days' average); (b) reanalysis driven case (REFC1SD, 1991-2010 summer days' average) minus the current climate online case (GCM2000, 2006-2025 summer days' average); (c) Future climate with current emission (GCM2100, 2106-2125 summer days' average) minus the current climate (GCM2000, 2006-2025 summer days' average); (d) Future climate with future emission (REFC2, 2080-2099 summer days' average) minus current climate (GCM2000, 2006-2025 summer days' average). Three black dots are representative stations (AREN1, SIPS1 and LAVO1).

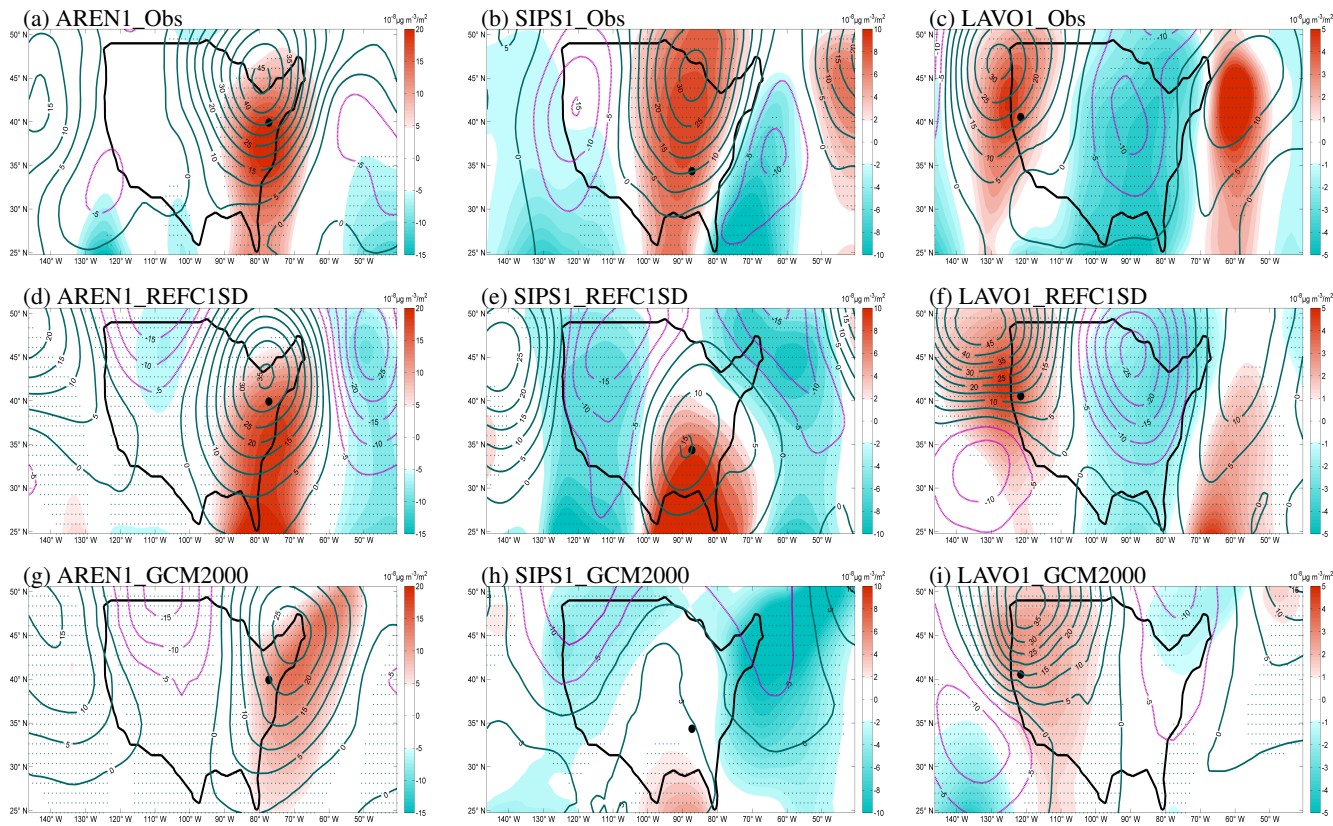

**Figure 4.** (Contour) composite 500 hPa geopotential height anomaly (positive values are represented by solid green lines and negative values by dashed magenta lines) and (shaded) regression coefficients between daily AWA and $PM_{2.5}$ at site (denoted by the black dots) (a, d, g) AREN1, (b, e, h) SIPS1 and (c, f, i) LAVO1 in the study domain for daily JJA time series of current climates. The top row are results using IMPROVE $PM_{2.5}$ and reanalysis AWA, the middle row uses the reanalysis driven simulated $PM_{2.5}$ (REFC1SD) and reanalysis AWA, and the bottom row uses current climate simulated $PM_{2.5}$ and AWA (GCM2000). Stippling indicates the regions that are statistically significant at the 5% confidence level. Unit: $10^{-8}$ $\mu$ g $m^{-3}$ $/m^2$ for regression coefficients.

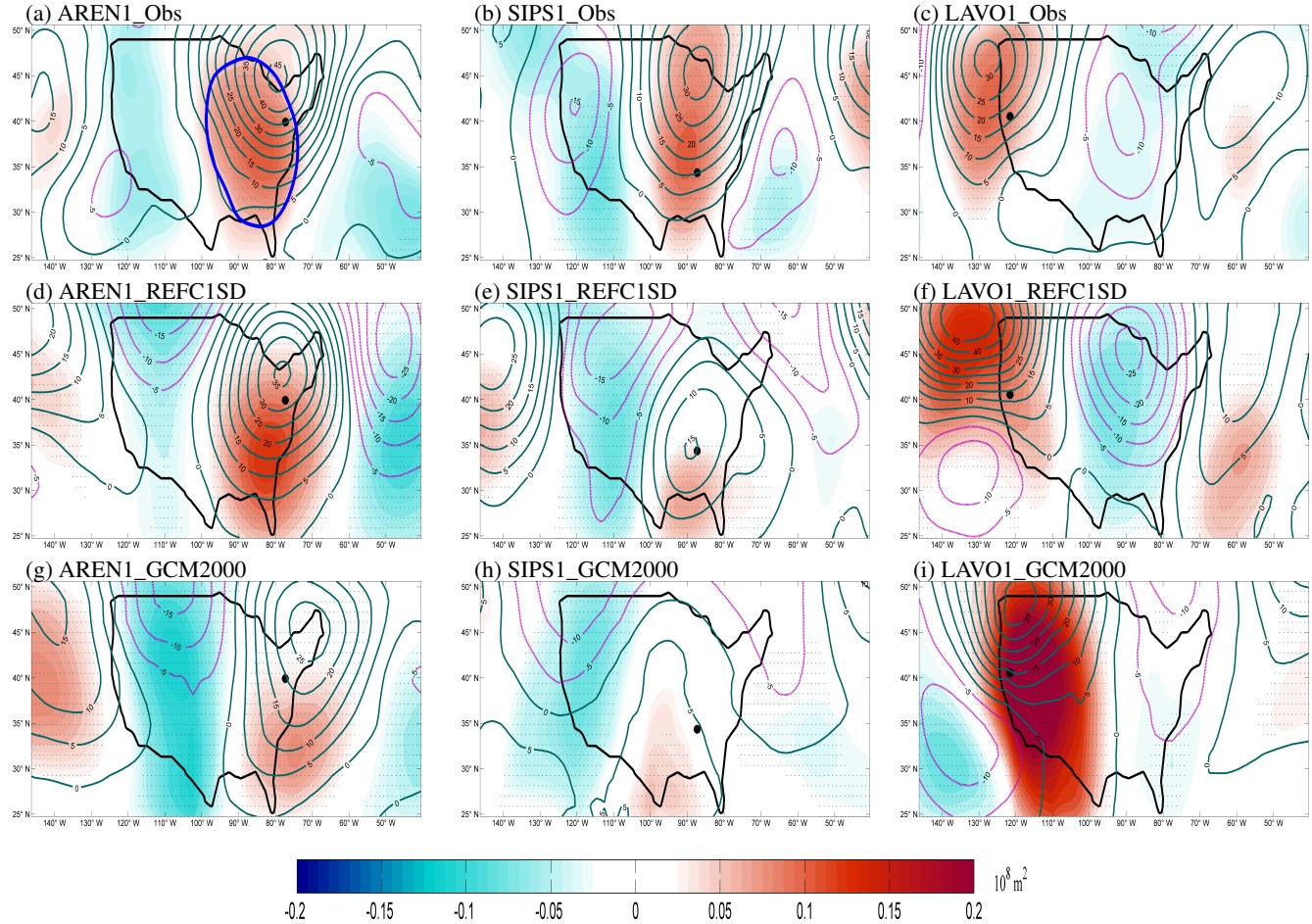

**Figure 5.** (Contour) composite 500 hPa geopotential height anomaly (positive values are represented by solid green lines and negative values by dashed magenta lines) and (shaded) corresponding AWA for PM$_{2.5}$ larger than $90^{th}$ quantile at site (denoted by the black dots) (a, d, g) AREN1, (b, e, h) SIPS1 and (c, f, i) LAVO1. The top row are results using IMPROVE PM$_{2.5}$ and reanalysis AWA, the middle row uses the reanalysis driven simulated PM$_{2.5}$ (REFC1SD) and reanalysis AWA, and the bottom row uses current climate simulated PM$_{2.5}$ and AWA (GCM2000). Stippling indicates the regions that are statistically significant at the 5% confidence level. Unit: m for 500 hPa geopotential height and $10^8$ $m^2$ for AWA. The blue outlined area in a) is the impact region, which is defined as the region of the maximum regression coefficient minus 0.05.

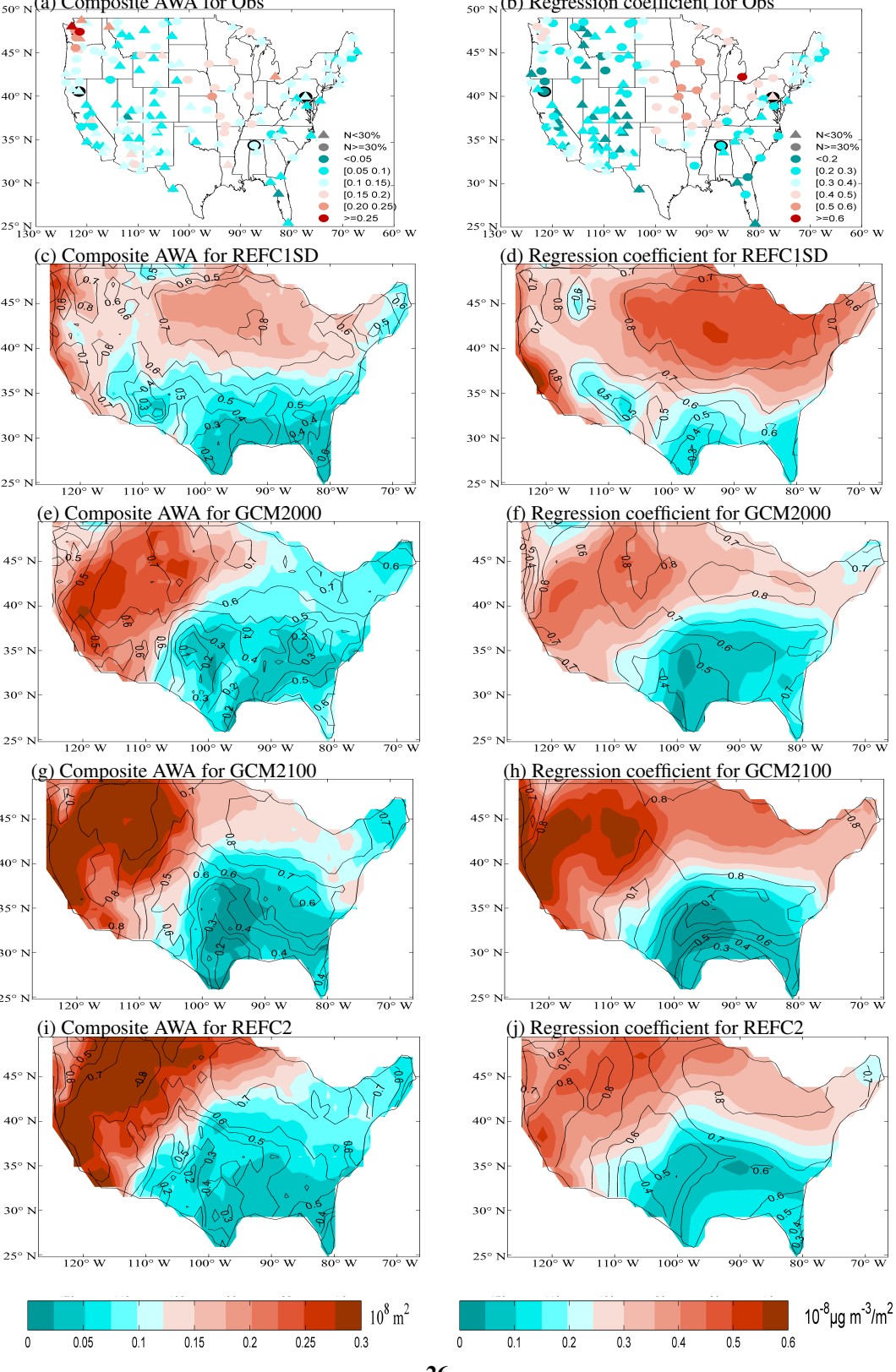

**Figure 6.** The maximum of the composite AWA distribution for $PM_{2.5}$ larger than $90^{th}$ quantile (shaded) (a, c, e, g, i), and the centers of the spatial regression coefficient distribution between $PM_{2.5}$ and AWA (b, d, f, h, j): observations (Obs, first row), current climate from the reanalysis driven simulation (REFC1SD, second row), current climate from the coupled model simulation (GCM2000, third row), future climate with current emission (GCM2100, fouth row) and future climate with future emission (REFC2, bottom row). At each grid point, the highest composite AWA anywhere in the domain based on the $PM_{2.5}$ larger than $90^{th}$ quantile and the highest regression coefficient between AWA and $PM_{2.5}$ are shown. In a) and b), the thee representative sites are denoted by the black dots. In a) and b) the different shapes (circle or triangle) indicate the number of values for every grid that are statistically significant (at the 5% confidence level) is more than 30% or not. The different colors indicate different highest composite AWA and regression coefficients as indicated in the legend. In c) through j) the number of values for every grid that are statistically significant at the 5% confidence level are shown (in black contours).

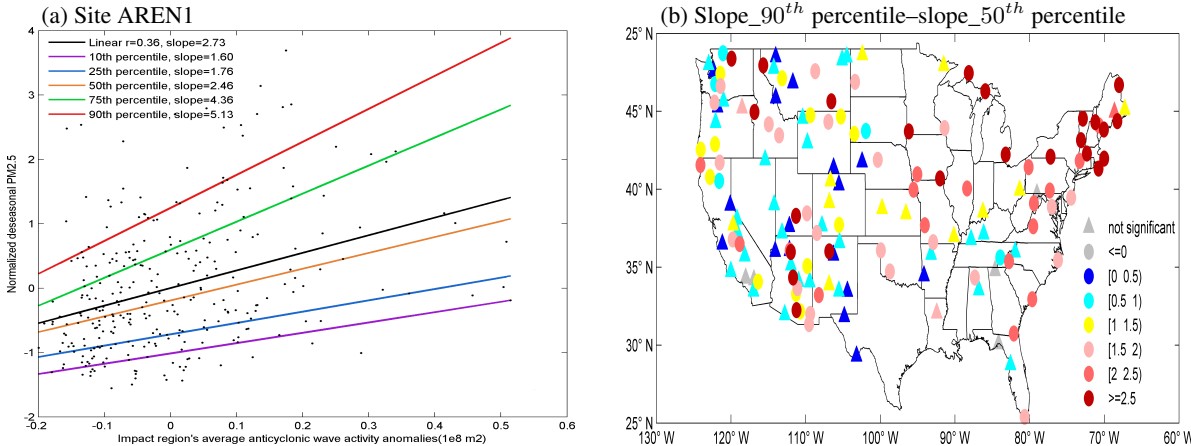

**Figure 7.** (a) Scatterplot for site AREN1 JJA deseasonalized $PM_{2.5}$ and impact region's average AWA with linear and quantile regression results. (b) The subtraction of $50^{th}$ percentile quantile regression slope from $90^{th}$ percentile quantile regression slope between $PM_{2.5}$ and impact region's average AWA across all 150 sites in the US (at the 5% significance level).

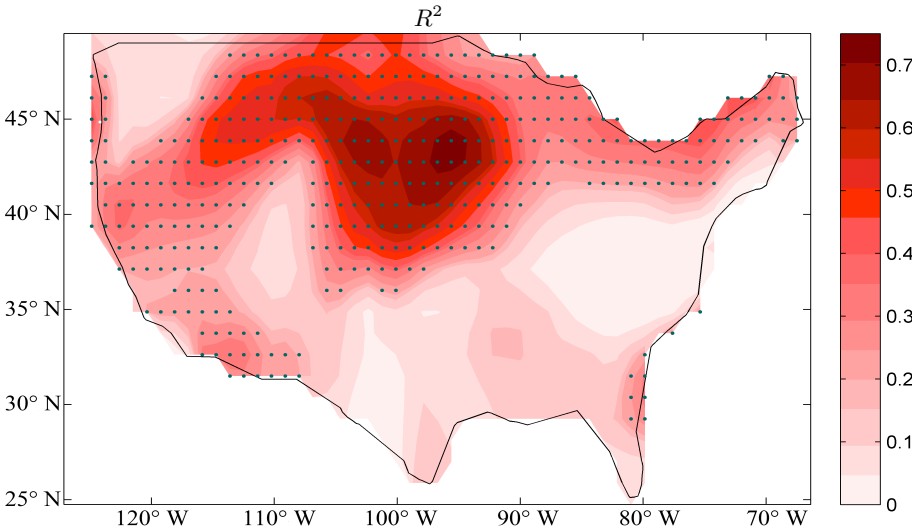

**Figure 8.** $PM_{2.5}$'s interannual variance explained ($R^2$) of the linear regression model using AWA projection value as the explanatory variable with modeled results. Stippling indicates where $R^2$ is significant (at the 5% significance level) at model grids.

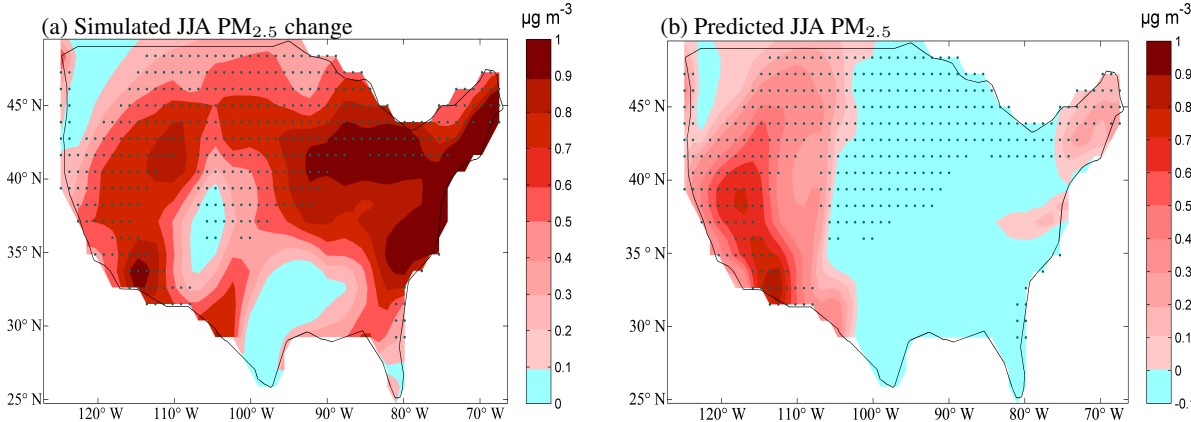

**Figure 9.** (a) Simulated change of JJA PM$_{2.5}$ between simulated future (future climate with current emission, GCM2100, 2106-2125 mean) and current (current climate from the coupled model simulation, GCM2000, 2006-2025 mean). (b) Predicted JJA PM$_{2.5}$ change using the linear regression model fitted with simulated current PM$_{2.5}$ (GCM2000, 2006-2025 mean). Stippling indicates where $R^2$ is significant (at the 5% significance level) at model grids. Unit: $\mu$ g $m^{-3}$ for PM$_{2.5}$.

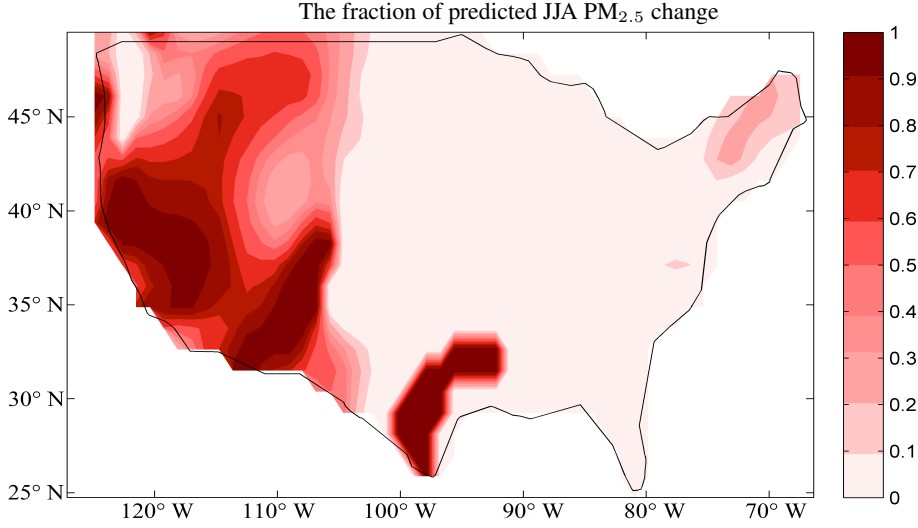

**Figure 10.** The fraction of predicted JJA PM$_{2.5}$ change using simulated data (current climate from the coupled model simulation, GCM2000, 2006-2025 mean; Figure 9b divided by Figure 9a) fitted model accounts for the total JJA PM$_{2.5}$ change from simulations. The fraction that less than zero is regarded as zero. Stippling indicates where $R^2$ is significant (at the 5% significance level) at model grids.