# Peer review of "The relationship between PM2.5 and anticyclonic wave activity during summer over the United States"

_Atmospheric Chemistry and Physics, 2021_

## Referee Comment (RC1)

**Comments on "The relationship between PM2.5 and anti-cyclone wave activity during summer over the United States" by Wang et al., 2021**

Based on regression analysis on anticyclone wave activity (AWA) anomalies and PM$_{2.5}$ concentrations, the present study attempts to evaluate the possible control of large-scale atmospheric circulation on regional aerosol pollution, which is very important for improving the understanding of air pollution formation and model prediction capability. Since the analysis was conducted on both observations and well-evaluated model simulations under various scenarios, the conclusions drawn from this study are generally reliable and robust. The manuscript is also well organized and prepared. Thus, I recommend it for publication with minor revision. Specific comments are listed as below.

Line 95: How to choose the three sites (i.e., AREN1, SIPS1, and LAVO1) as the representative stations of different part of the country and why? For example, the authors may want to provide more detailed information regarding the representativeness of the three selected stations.

Line 222: It seems that the statement here flipped two simulation terms: it is REFC1SD which is for the reanalysis driven simulations and GCM2000 for the coupled model simulations, right? I also found several other places such as in figure captions show similar typos. Please go through the entire paper and make sure that the case terms are not messed up.

Lines 295-296: why Wise and Comrie (2005) show much lower coefficient of determinations (i.e., 0.1-0.5) than this study (i.e., 0.75)?

Lines 284-286: how about the comparison of PM2.5 results in this study with that in Porter et al. (2015)? Are they consistent with each other as what is shown in ozone?

Line 150: the full term of AWA should be shown at its first instance (i.e., in Line 59).

Section 2.3: Since the AWA (or LWA) is the key variable in this study, it's better to list the equation(s) used to calculate AWA (or LWA) so that the readers don't need to refer to previous references to understand the detailed calculations related to AWA (or LWA).

Lines 225-226: miss commas after "In addition" and before "suggesting".

Figures & Table:

Fig. 4: please describe what the contour lines (green and magenta) stand for.

Fig. 6 caption: GCM2100 should be the case with the future climate with current emission while REFC2 is the future climate with current emission. The original description seems wrong based on methodology section.

Fig. 9: Is the panel (a) for the change between GCM2100 and GCM2000 or between GCM2100 and REFC2 simulation? I note that REFC2 is for future climate as denoted in methodology section, right? Similar issue in Fig. 10 caption.

Table 1: The time period information for GCM2000 and GCM2100 do not mean anything, as you just simulated a climatology, not specific years. It should be sufficient to just mention the length of the simulations.

---

## Author Comment (AC1)

Responses to reviewer 1
We would like to thank the reviewer for the positive and constructive comments about the manuscript (The relationship between PM$_{2.5}$ and anti-cyclone wave activity during summer over the United States). We have studied the comments carefully and made corrections. The corrections are listed in the following responses and all are completed and marked in the manuscript in yellow.

**Comments on "The relationship between PM2.5 and anti-cyclone wave activity during summer over the United States" by Wang et al., 2021**

Based on regression analysis on anticyclone wave activity (AWA) anomalies and PM2.5 concentrations, the present study attempts to evaluate the possible control of large-scale atmospheric circulation on regional aerosol pollution, which is very important for improving the understanding of air pollution formation and model prediction capability. Since the analysis was conducted on both observations and well-evaluated model simulations under various scenarios, the conclusions drawn from this study are generally reliable and robust. The manuscript is also well organized and prepared. Thus, I recommend it for publication with minor revision. Specific comments are listed as below.

Line 95: How to choose the three sites (i.e., AREN1, SIPS1, and LAVO1) as the representative stations of different part of the country and why? For example, the authors may want to provide more detailed information regarding the representativeness of the three selected stations.

Response: There are two reasons for the representativeness of the three selected stations.

(1) The three sites here correspond to the three sites in Northeast, Southeast and Western region respectively, which has differing impacts of meteorological persistence on the distribution and extremes of ozone in Sun et al. (2017).

(a) AREN1 (39.92°N, 77.31°W) matches with the site PSU106 (40.72°N, 77.93°W, in the Northeast region) which has a well-known association between high ozone and stagnation.

(b) SIPS1 (34.34°N, 87.34°W) matches with the site SND152 (34.29°N, 85.97°W, in the Southeast region) which is least sensitive to the length of a stagnation event for ozone in the Southeast (ozone increases by ~0.06 standard deviation per day on average).

(c) LAVO1 (40.54°N, 121.58°W) matches with the site LAV410 (40.54°N, 121.58°W, in the Western region) which is noted for the fewest number of days between cyclones of 4 days or longer. Furthermore, LAVO1 is considered to be a clean air site in California where anthropogenic influence is at a minimum (Vancure et al., 2002). LAVO1 is a higher elevation (1.76 km) site in Northern California that has also been used to quantify baseline ozone concentrations due to its relatively isolated location (Parrish et al., 2012). In addition, the long range transport from Asia and meteorology are dominant drivers of pollutants at LAVO1 by distinguishing among local, distant North American, and Asian sources of particulate matter (PM$_{2.5}$) and O$_3$ (Vancure et al., 2015).

The more detailed information regarding the representativeness of the three selected stations are included in lines 97-106 as follows:

We chose three representative stations in different parts of the country to investigate the relation between AWA and PM$_{2.5}$ in detail. The IMPROVE station names are AREN1 (Arendtsville, Pennsylvania; 39.92 °N, 77.31°W; in the Northeast), SIPS1 (Sipsey Wilderness, Alabama; 34.34°N, 87.34°W; in the Southeast) and LAVO1 (Lassen Volcanic NP, California; 40.54°N, 121.58°W; in the West), which are shown with the red dots in Figure 1. They match with the site PSU106 (40.72°N, 77.93°W, in the Northeast), SND152 (34.29°N, 85.97°W, in the Southeast) and LAV410 (40.54°N, 121.58°W, in the West) respectively, which has differing impacts of meteorological persistence on

the distribution and extremes of ozone in Sun et al. (2017) to allow comparison between the ozone and PM$_{2.5}$ response to AWA. Long range transport from Asia and meteorology are dominant drivers of pollutants at LAVO1, where anthropogenic influence is at a minimum as a clean air site in California (Vancure et al., 2015).

(2) The three sites of different part of the country differ from each other climatologically (Figure 2). The climatological average for PM$_{2.5}$ is greater in the Eastern than in the Western sites. The highest correlation coefficients between model and observations (0.93) are seen at SIPS1 perhaps due to the large seasonal variation in PM$_{2.5}$ concentrations. Emission changes are more important than climate changes at AREN1, but it is not clear which is more important at SIPS1 or LAVO1.

| | This study | Wise and Comrie (2005) |
|---|---|---|
| object | $PM_{2.5}$ and AWA | PM and meteorological variables |
| period | current (2006-2025); future (2106-2125) | 1990-2003 |
| data | the IMPROVE data and CESM simulations | Meteorological data were obtained from the National Climatic Data Center and collected by the standard protocols established by the US National Weather Service. |
| study area | 150 IMPROVE sites across US | the Southwest's five major metropolitan areas |
| method | univariate linear regression | stepwise regression; the KZ filter method |
| results | Future changes in US $PM_{2.5}$ based only on changes in climate are estimated to increase $PM_{2.5}$ | Moisture levels (particularly relative humidity) are the strongest predictors of PM concentrations in all five cities examined. |

| | concentrations due to increased AWA in summer over areas where PM2.5 variations are dominated by meteorological changes, especially over the western US. Changes between current and future climates in AWA can explain up to 75% of $PM_{2.5}$ variability using a linear regression model. | Meteorological variability typically accounts for 20–50% of PM variability. Long-term trends in PM concentrations were relatively flat in all five cities analyzed but contained coincident extremes unrelated to meteorology. |
|---|---|---|

(4) Because of all the differences in the methodologies, it is unclear what the underlying differences are. We add the following sentence in lines 319-321:

Wise and Comrie (2005) similarly determined $R^2$ values of 0.1-0.5 for associations of PM with atmospheric variables across sites in the Southwest, and here we see a comparable relationships across the Southwest, although these studies use different methodologies as well as consider different time periods.

Lines 284-286: how about the comparison of PM2.5 results in this study with that in Porter et al. (2015)? Are they consistent with each other as what is shown in ozone?

Response:

Porter et al. (2015) find nationally averaged sensitivities of 95th percentile summer $O_3$ to changes in maximum daily temperature of approximately 0.9 ppb $°C^{-1}$, while the sensitivity of 50th percentile summer $O_3$ (the annual median) is only 0.6 ppb$°C^{-1}$. They also obtain the greater sensitivities of $PM_{2.5}$ at the highest concentration percentiles to mean daily temperature. While the sensitivities of $O_3$ to temperature are the greatest along both the northeast coast and Southern California, $PM_{2.5}$ sensitivities to temperature peak entirely in the east due to the regionality of $PM_{2.5}$ speciation. Out of the 150 sites, 145 sites in this study show that 90th percentile $PM_{2.5}$ increases more than the 50th percentile of $PM_{2.5}$ with the enhancement of the AWA. In the Northeast region (north and east of New York state with New York state included), this relationship is the most pronounced. This difference in response between the highest and median $PM_{2.5}$ values indicates the different sensitivities within various percentiles of the $PM_{2.5}$ levels. These results are to some extent consistent with those from Porter et al. (2015), which addressed that averaged sensitivity of 95th percentile summertime ozone to changes in highest daily temperature was larger than the sensitivity of 50th percentile summertime ozone.

The comparison of $PM_{2.5}$ results in this study with that in Porter et al. (2015) is updated to focus on $PM_{2.5}$ in lines 308-310 as follows:

These results are to some extent consistent with those from Porter et al. (2015), which addressed the greater sensitivities to mean daily temperature at the highest concentration percentiles in predicting summertime $PM_{2.5}$, but with $PM_{2.5}$ sensitivities to temperature peak entirely in the east due to the regionality of $PM_{2.5}$ speciation.

Line 150: the full term of AWA should be shown at its first instance (i.e., in Line 59).
Response: As suggested, the full term of AWA (anti-cyclone wave activity) is shown at its first

instance in line 2.

Section 2.3: Since the AWA (or LWA) is the key variable in this study, it's better to list the equation(s) used to calculate AWA (or LWA) so that the readers don't need to refer to previous references to understand the detailed calculations related to AWA (or LWA).

Response: As suggested the calculation for the AWA is included in lines 151-160 in section 2.3 as follows:

To calculate AWA, we adopt the procedures in Chen et al. (2015) and Huang and Nakamura (2016). A dynamical quantity, q (here we use $Z_{500}$, geopotential height at 500 hPa), approximately decreases with latitude in the Northern Hemisphere. For a given value of q = Q, we introduce an equivalent latitude $\emptyset_e(Q)$ as

$$\emptyset_e(Q) = \sin^{-1}[1 - \frac{S(Q)}{2\pi a^2}]$$

Here, S(Q) is the area bounded by the Q contour towards the North Pole and a denotes Earth's radius. Defining an eddy term as $\hat{q} \equiv q - Q$ and separating the southward and northward displacements in the Q contour, we calculate the cyclonic (southern), anticyclonic (northern) and total LWA at the longitude $\lambda$ and latitude $\emptyset_e$ by

$$A_C(\lambda, \emptyset_e) = \frac{a}{\cos\emptyset_e} \int_{\hat{q}\leq0,\emptyset\leq\emptyset_e(Q),\lambda=cons}^{\blacksquare} \hat{q} \cos\emptyset d\emptyset$$

$$A_A(\lambda, \emptyset_e) = \frac{a}{\cos\emptyset_e} \int_{\hat{q}\geq0,\emptyset\geq\emptyset_e(Q),\lambda=const}^{\blacksquare} \hat{q} \cos\emptyset d\emptyset$$

$$A_T(\lambda, \emptyset_e) = A_C - A_A$$

More details on LWA theory and derivation can be found in Chen et al. (2015) and Huang and Nakamura (2016).

Lines 225-226: miss commas after "In addition" and before "suggesting".

Response: Commas are included after "In addition" and before "suggesting" in lines 246-247.

Figures & Table:

Fig. 4: please describe what the contour lines (green and magenta) stand for.

Response: Thank you for pointing this out: the contour lines (green and magenta) is described in caption for Fig4 as follows:

Figure 4. (Contour) composite 500 hPa geopotential height anomaly (positive values are represented by solid green lines and negative values by dashed magenta lines) and (Shaded) regression coefficients between daily AWA and $PM_{2.5}$ at site (denoted by the black dots) (a, d, g) AREN1, (b, e, h) SIPS1 and (c, f, i) LAVO1 in the study domain for daily JJA time series of current climates. The top row are results using IMPROVE $PM_{2.5}$ and reanalysis AWA, the middle row uses the reanalysis driven simulated $PM_{2.5}$ (REFC1SD) and reanalysis AWA, and the bottom row uses current climate simulated $PM_{2.5}$ and AWA (GCM2000). Stippling indicates the regions that are statistically significant at the 95% confidence level. Unit: $10^{-8} \mu g \, m^{-3} /m^2$ for regression coefficients.

Fig. 6 caption: GCM2100 should be the case with the future climate with current emission while REFC2 is the future climate with current emission. The original description seems wrong based on methodology section.

Response: Thank you for pointing this out: GCM2100 is the future climate with current emission and REFC2 is the future climate with future emission. The description for REFC2 is wrong here and it is revised as follows:

The maximum of the composite AWA distribution for $PM_{2.5}$ larger than $90^{th}$ quantile(Shading) (a, c, e, g, i), and the centers of the spatial regression coefficient distribution between $PM_{2.5}$ and AWA (b, d, f, h, j): observations (Obs, first row), current climate from the reanalysis driven simulation (REFC1SD, second row), current climate from the coupled model simulation (GCM2000, third row), **future climate with current emission (GCM2100, fouth row) and future climate with future emission (REFC2, bottom row)**. At each grid point, the highest composite AWA anywhere in the domain based on the $PM_{2.5}$ larger than $90^{th}$ quantile and the highest regression coefficient between AWA and $PM_{2.5}$ are shown. In a) and b), the thee representative sites are denoted by the black dots. In a) and b) the different shapes (circle or triangle) indicate the number of values for every grid that are statistically significant (at the 5% confidence level) is more than 30% or not. The different colors indicate different highest composite AWA and regression coefficients as indicated in the legend. In c) through j) the number of values for every grid that are statistically significant at the 5% confidence level are shown (in black contours).

Fig. 9: Is the panel (a) for the change between GCM2100 and GCM2000 or between GCM2100 and REFC2 simulation? I note that REFC2 is for future climate as denoted in methodology section, right? Similar issue in Fig. 10 caption.

Response: We agree that the descriptions for Figure 9 and Figure 10 are not clear enough. They are revised as follows:

Figure 9. (a) Simulated change of JJA $PM_{2.5}$ between simulated future (future climate with current emission, GCM2100, 2106-2125 mean) and current (current climate from the coupled model simulation, GCM2000, 2006-2025 mean). (b) Predicted JJA $PM_{2.5}$ change using the linear regression model fitted with simulated current PM2.5 (GCM2000, 2006-2025 mean). Stippling indicates where $R^2$ is significant (at 5% significance level) at model grids. Unit: μ g m$^{-3}$ for $PM_{2.5}$.

Figure 10. The fraction of predicted JJA $PM_{2.5}$ change using simulated data (current climate from the coupled model simulation, GCM2000, 2006-2025 mean; Figure 9b divided by Figure 9a) fitted model accounts for the total JJA $PM_{2.5}$ change from simulations. The fraction that less than zero is regarded as zero.

Table 1: The time period information for GCM2000 and GCM2100 do not mean anything, as you just simulated a climatology, not specific years. It should be sufficient to just mention the length of the simulations.

Response: Good point: the time period is removed and the table 1 is revised as follows:

| Simulation (years) | Model Case Name | $GHG^1$ forcing | $SST^2$ and sea ice | Emissions | Meterological driver |
|---|---|---|---|---|---|
| REFC1SD (1991-2010) | f.e11.TSREFC1SD.f19. f19.ccmi23.001 | $CMIP5^3$ | $HadISST2^4$ | Anthropogenic and biomass burning emission: $MACCity^5$ Biogenic emissions: $MEGAN2^6$ | $MERRA^7$ |
| GCM2000 (2006-2025) | b.e11.TSREFC2.femis 2000.y2000.f19.f19. ccmi23.001 | $CO_2 = 369$ ppm | $Online^8$ | Anthropogenic and biomass burning from $AR5^9$ Biogenic emissions: Monthly values from MEGAN2 for 2000 | Online |
| GCM2100 (2106-2125) | b.e11.TSREFC2.femis 2000.y2100.f19.f19. ccmi23.001 | $CO_2 = 669$ ppm | $Online^8$ | Same as GCM2000 | $Online^8$ |
| REFC2 (2080-2099) | b.e11.TSREFC2.f19.g1 6.ccmi23.001.cam.h7 | A1B scenario | Online | A1B scenario | Online |

---

## Author Comment (AC2)

In this manuscript the authors explore one aspect of the known connection between meteorology and pollution, focusing on statistical correlations between anti-cyclone wave activity (AWA) and fine particulate matter (PM2.5). This is a useful topic, and one which has attracted a great deal of attention in recent years in the hopes of better understanding variability in observed pollution levels as well as improving and constraining projections of future climate impacts on pollution. While I see some promising steps and interesting results here, I also see serious omissions that make it hard for me to understand and accept the final conclusions being drawn. I would like to see significant improvement and clarification in several areas before I can recommend publication in ACP.

Response:We would like to thank the reviewer for the positive and constructive comments about the manuscript. The improper expressions and figures that the reviewer mentioned have been carefully revised. The updates are listed in the following responses and all are completed in the manuscript in yellow.

General questions and comments

PM2.5 speciation

Unlike ozone, previously explored by the authors using a similar methodology, PM2.5 is composed of a wide array of compounds derived from many different precursors. The treatment of PM2.5 here, though, seems to completely ignore this detail and handles all particles as identical. Without any individual treatment of speciated PM2.5, or even discussion of how variability in composition across the United States may be affecting relevant formation/loss mechanisms or final correlations, it's very difficult to accept any of the results or conclusions. This aspect of the analysis needs serious work to address this dimension.

Response:

We acknowledge your point. We don't currently have the output on the speciation. There are not many measurements of the $PM_{2.5}$ speciation as correct as just of $PM_{2.5}$ itself. Furthermore, $PM_{2.5}$ is regulated quantity in the US with health implications. Considering these points, this study focus on the importance of the anti-cyclone wave activity in $PM_{2.5}$ changes across the US and the role of anti-cyclone wave activity in predicting future $PM_{2.5}$. How major components of $PM_{2.5}$ react to wave activity using IMPROVE and available model output could be studied in the future, including the discussion of how variability in composition affecting relevant formation/loss mechanisms or final correlations across the US.

PM2.5/AWA mechanisms and correlation with other local meteorological metrics

While at times it is mentioned that AWA has ties to local meteorological variables such as temperature and stagnancy, these correlations are never shown, quantified, nor discussed in any detail, and their potential covariance is crucial in terms of explaining the mechanisms underpinning the presented relationships. If it is being argued that variability in AWA can explain "up to 75% of interannual PM2.5 variability across the US", it behooves the authors to go into greater detail on the exact mechanisms from which this predictive power stems. Are higher temperature driving increased emissions? Is it transport related? Absence of wet deposition? As it stands this open question is the elephant in the room of this manuscript's discussion and conclusions, and needs to be addressed (ideally with conclusive and quantitative analytical tools).

Response:

There is a substantial literature between AWA and local meteorology. The connections between AWA and local meteorology are included in Introduction section (in lines 55-59) as follows:

In addition, temperature extremes are more likely to connect with wave events through the large-amplitude troughs and ridges (Pfahl and Wernli, 2012; Chen et al., 2015; Martineau et al., 2017). Shen and Mickley (2017) identify the association between warm tropical Atlantic sea surface temperatures and enhanced subsidence, reduced precipitation and increased temperatures through stationary wave propagation in the Eastern US.

We agree and add more information on the AWA relationship with meteorology. Future changes (GCM2100-GCM2000) in JJA 500 hPa geopotential height anomaly are shown to explain the interannual $PM_{2.5}$ variability resulting from variability in AWA. Future changes in JJA geopotential height anomaly at 500 hPa between the future period (GCM2100 simulation) relative to the current period (GCM2000 simulation) are shown in Figure S1. The 500 hPa geopotential height increases everywhere by approximately 25–55 m over the entire US. These strengthened geopotential height can be explained by mid-to-high latitude warming in the future climate. The increased geopotential height at higher latitudes is consistent with other model projections (Yue et al., 2015; Vavrus et al., 2017). The change in 500 hPa height which shows a distinct anticyclonic pattern centering over the western US and the adjacent ocean is consistent with changes in a suite of atmospheric variables related to regional air quality (Meehl et al., 2004). These conditions include warmer temperature (2–4 ℃), more downward solar radiation (up to 15 W/m$^2$), less frequent rainfall (up to 8 days less per season), more frequent stagnation (up to 15 days more per season), and reduced ventilation (up to 2 more unvented hours per day) (Leung et al., 2005). Changing these meteorological parameters strongly affect changes in $PM_{2.5}$ concentrations (Porter et al., 2015). The spatially variable response of total $PM_{2.5}$ to temperature changes reflected the combined responses of nitrate and sulfate, while the simulation-long average $PM_{2.5}$ concentration over land grid cells decreased with increasing wind speed by 50 ngm$^{-3}$ %$^{-1}$ (0.77%%$^{-1}$) and increased with water vapor concentration by 20 ngm$^{-3}$ %$^{-1}$ (0.29%%$^{-1}$) in July (Dawson et al., 2007). In addition, $PM_{2.5}$ concentrations are on average 2.6 μg m$^{-3}$ higher on stagnant vs. non-stagnant days (Tai et al., 2010).

[Figure]

A short explanation for the $PM_{2.5}$ variability resulting from changes in AWA is included in lines 346-352 in section 3.4 as follows:

Figure S2. JJA-simulated difference for 500 hPa geopotential height (m) between the GCM2100 and the GCM2000.

References

Chen, G., Lu, J., Burrows, D. A., and Leung, L. R.: Local finite amplitude wave activity as an objective diagnostic of midlatitude extreme weather, Geophys. Res. Lett., 42, 10952–10960, https://doi.org/10.1002/2015GL066959, 2015.

Dawson, J. P., Adams, P. J., and Pandis, S. N.: Sensitivity of PM2.5 to climate in the Eastern US: a modeling case study, Atmos. Chem. Phys., 7 (16), 4295-4309, 2007.

Leung, L. R. and Gustafson, Jr., W. I.: Potential regional climate change and implications to U.S. air quality, Geophys. Res. Lett., 32, L16711, doi:10.1029/2005GL022911, 2015.

Martineau, P., Chen, G., and Burrows, D. A.: Wave events: climatology, trends, and relationship to Northern Hemisphere winter blocking and weather extremes, J. Climate, 30, 5675–5697, 2017.

Meehl, G. A. and Tebaldi, C.: More intense, more frequent, and long lasting heat waves in the 21st century, Science, 305, 994– 997, 2004.

Pfahl, S., and Wernli H.: Quantifying the relevance of atmospheric blocking for co-located temperature extremes in the Northern Hemisphere on (sub-)daily time scales, Geophys. Res. Lett., 39, L12807, doi:10.1029/2012GL052261, 2012.

Porter, W. C., Heald, C. L., Cooley, D., and Russell, B.: Investigating the Observed Sensitivities of Air-Quality Extremes to Meteorological Drivers via Quantile Regression, Atmos. Chem. Phys., 15(18), 10349–10366, doi: 10.5194/acp-15-10349- 2015, 2015.

Shen, L. and Mickley, L. J.: Seasonal prediction of US summertime ozone using statistical analysis of large scale climate patterns, P. Natl. Acad. Sci. USA, 114, 2491–2496, 2017.

Tai, A. P. K., Mickley, L. J., and Jacob, D. J.: Correlations between fine particulate matter ($PM_{2.5}$) and meteorological variables in the United States: Implications for the sensitivity of $PM_{2.5}$ to climate change, Atmos. Environ., 44, 3976–3984, 2010.

Vavrus, S. J., Wang, F., Martin, J. E., Francis, J. A., Peings, Y., and Cattiaux, J.: Changes in North American atmospheric circulation and extreme weather: Influence of Arctic amplification and Northern Hemisphere snow cover, J. Climate, 30, 4317–4333, 2017.

Yue, X., Mickley, L. J., Logan, J. A., Hudman, R. C., Martin, M. V., and Yantosca, R. M.: Impact of 2050 climate change on North American wildfire: consequences for ozone air quality, Atmos. Chem. Phys., 15, 10033–10055, https://doi.org/10.5194/acp-15-10033-2015, 2015.

Specific section questions and comments

Section 2.1: Unless I am misunderstanding some key detail, 1000 observations between 1988 and 2014 seems like a very low threshold for data availability, with potentially very significant biases due to missing data. Were these observations counted year round, or only from summers? Were there any notable biases or differences between selected stations in terms of whether the observations occurred early or late in the temporal domain? Also, why does this year range (1988-2014) differ from the date ranges used in other data sources (for example 1991-2010 for ERA-Interim data)?

Response:

For this study we focus on data with longer time periods, and just for the summer time. In order to make the results robust, those sites with short observational period are omitted. A total of 233 observational sites starting from 1988 for $PM_{2.5}$ is in IMPROVE (http://views.cira.colostate.edu/fed/QueryWizard/Default.aspx). Only those sites with observational period less than 6 years are omitted, such as AMBL1 (starting from 2003 and ending 2004), BALT1(starting from 2004 and ending 2006) and DOLA1 (starting from 1994 and ending 1999) and so on. In order to avoid biases, all stations for summer (June, July, August, JJA hereafter) with at least 1000 valid $PM_{2.5}$ values between 1988 and 2014 are collected in this study, totaling 150 stations. Only 28 observational data of 150 starts from 1988 and ends 2014. 40 observational data of 150 stations starts between 1990 and 1999. 81 observational data of 150 stations starts between 2000 and 2009. In order to avoid biases by including observational data as much as possible and covering domain as large as possible, the data period between 1988 and 2014 is selected.

The descriptions of the 150 observational sites used in this study are included in the online supplement Table S1.

Table S1. The descriptions of the 150 observational sites used in this study are included in the online supplement.

| Site | Code | State | Latitude | Longitude | Elevat | tartDate | ndDate | Number of ob |
|------|------|-------|----------|-----------|--------|----------|--------|--------------|
| Acadia NP | ACAD1 | ME | 44.38 | −68.26 | 157 | 198803 | 201412 | 3050 |
| Addison Pinnacl | ADPI1 | NY | 42.09 | −77.21 | 512 | 200104 | 201006 | 1125 |
| Agua Tibia | AGTI1 | CA | 33.46 | −116.97 | 507 | 200011 | 201412 | 1709 |
| Arendtsville | AREN1 | PA | 39.92 | −77.31 | 267 | 200104 | 201012 | 1187 |
| Badlands NP | BADL1 | SD | 43.74 | −101.94 | 736 | 198803 | 201412 | 3050 |
| Mount Baldy | BALD1 | AZ | 34.06 | −109.44 | 2508 | 200002 | 201412 | 1798 |
| Bandelier NM | BAND1 | NM | 35.78 | −106.27 | 1988 | 198803 | 201412 | 3051 |
| Big Bend NP | BIBE1 | TX | 29.30 | −103.18 | 1066 | 198803 | 201412 | 3051 |
| North Birmingha | BIRM1 | AL | 33.55 | −86.81 | 175 | 200404 | 201412 | 1320 |

Section 2.2: I'm concerned about the inclusion and comparison between free running and specified dynamics model output, as these two types can differ in very significant ways, even for otherwise identical years. In particular, Figure 3 (and related analysis) shows a very strong difference between REFC1SD and GCM2000. How much of that difference could be an artifact of the differing dynamics? Do the overlapping years between them (2006-2010) show good agreement in AWA? If not, this has major consequences for the interpretation of 3b and 3d.

Response:

The difference between free-running models and reanalyses-forced is very interesting, and explicitly considered in this paper. The model years in the GCM2000 are arbitrary (and the dates removed in the new version, as per the request of the other author, so other readers are not similarly confused), since the model is free running, and should not be considered equivalent to 2006-2010. The reanalyses-forced runs will hopefully do a better job matching the interannual variability, but cannot be used as a control for a future climate run, so we include a current climate control run to use for comparison with the future. Because we consider 20 years the climatologies can be compared however, which is why we use long time periods. Thus it is possible to contrast the free running model and the reanalyses in figures 3b and 3d.

Section 2.6: This description needs some serious clarification. I was not able to clearly understand this description of "composite methodology" until piecing it together from later sections and figures.

Response:

The characteristics and references for the composite methodology are included in lines 205-208 in section 2.6 as follows:

We use a composite methodology which is based around the most polluted (>90th percentile) daily $PM_{2.5}$ and the corresponding anomalies at every station. Composite 500 hPa geopotential height and AWA for daily values of $PM_{2.5}$ larger than 90th percentile are produced by separately averaging all daily anomaly values of the corresponding 500 hPa geopotential height and AWA. The composite methodology can average out much of the variability. Composite Madden–Julian Oscillation cycles of precipitation and ozone for each phase are examined by averaging together all daily anomaly values for the given quantity separately (Sun et al., 2014). In addition, the meteorological conditions conductive to a high ozone event are investigated by compositing about the first day of each high ozone event in the Northeastern region (Sun et al., 2017).

References

Sun, W., Hess, P., and Tian, B.: The response of the equatorial tropospheric ozone to the Madden–Julian Oscillation in TES satellite observations and CAM-chem model simulation, Atmos. Chem. Phys., 14, 11775–11790, 2014.

Sun, W., Hess, P., and Liu, C.: The impact of meteorological persistence on the distribution and extremes of ozone, Geophys. Res. Lett., 44, 1545–1553, 2017.

Figure 2: With exclusive focus on summer conditions in this paper, what is the relevance of including full annual cycles of PM2.5? Furthermore, what is the relevance of agreement between monthly average values year round (9 months of which are completely ignored in the rest of the paper) rather than daily values within the chosen JJA domain? Far more useful, (and legible) in my view, would be box or column plots showing summertime means/ranges for each of the different sources of data or model output, with correlation statistics provided for daily comparison to observations.

Response:

Although boxplot can display the spatial and temporal $PM_{2.5}$ data variations by visualizing much

information such as the maximum, upper quantile, median, lower quantile and minimum, Figure 2 can set the summer values within a broader context. There are some reasons for selecting Figure 2 in this study. Initially, the full annual cycle of $PM_{2.5}$ is used to show that $PM_{2.5}$ concentrations are largest during summer across months at the three sites. So that we can focus on summer months in this study. Furthermore, the significant correlation between observations and simulations is demonstrated using Figure 2. A statistically significant correlation (r>0.80, p<0.01; Figure 2) for current $PM_{2.5}$ is found between observations and simulations of monthly mean climatological averages (REFC1SD and GCM2000) at three representative sites. The highest correlation coefficients between model and observations (0.93) are seen at SIPS1 perhaps due to the large seasonal variation in $PM_{2.5}$ concentrations (Figure 2b). Finally, the correlation coefficients between future and current $PM_{2.5}$ is demonstrated using Figure 2. The future $PM_{2.5}$ concentrations are increased in GCM2100 under current emissions compared with current climate $PM_{2.5}$ simulations. There is a strong decease in climatological mean for future $PM_{2.5}$ at AREN1 under future emissions and meteorology (REFC2), while the climatological average for future $PM_{2.5}$ has no significant change under future emissions at SIPS1 and LAVO1. Such differences in the monthly mean averages for $PM_{2.5}$ suggests that emission changes are more important than climate changes at AREN1, but it is not clear which is more important at SIPS1 or LAVO1.

Section 3, site selection: The LAVO1 site seems like a problematic choice, relative to the other two. Not only is PM2.5 not particularly high during the summer, there appears to be a very high degree of variability during those months. Is there a reason for its selection? Also, considering the final conclusions focused on the importance of the AWA in the Midwest and Great Plains regions, wouldn't it be beneficial to have at least one site within one or both of them?
Response:
   There are two reasons for the representativeness of the three selected stations.
   (1) In order to make the results robust, the three representative sites are from the reference (Sun et al., 2017). The three sites here correspond to the three sites in Northeast, Southeast and Western region respectively, which has differing impacts of meteorological persistence on the distribution and extremes of ozone in Sun et al. (2017) to allow comparison to that study here. AREN1 (39.92°N, 77.31°W) matches with the site PSU106 (40.72°N, 77.93°W, in the Northeast region) which has a well-known association between high ozone and stagnation. SIPS1 (34.34°N, 87.34°W) matches with the site SND152 (34.29°N, 85.97°W, in the Southeast region) which is least sensitive to the length of a stagnation event for ozone in the Southeast (ozone increases by ~0.06 standard deviation per day on average). LAVO1 (40.54°N, 121.58°W) matches with the site LAV410 (40.54°N, 121.58°W, in the Western region) which is noted for the fewest number of days between cyclones of 4 days or longer. Furthermore, LAVO1 is considered to be a clean air site in California where anthropogenic influence is at a minimum (Vancure et al., 2002). LAVO1 is a higher elevation (1.76 km) site in Northern California that has also been used to quantify baseline ozone concentrations due to its relatively isolated location (Parrish et al., 2012). In addition, the long range transport from Asia and meteorology are dominant drivers of pollutants at LAVO1 by distinguishing among local, distant North American, and Asian sources of particulate matter ($PM_{2.5}$) and $O_3$ (Vancure et al., 2015).
   (2) The three sites of different part of the country differ from each other climatologically (Figure 2). The climatological average for $PM_{2.5}$ is greater in the Eastern than in the Western sites. The

highest correlation coefficients between model and observations (0.93) are seen at SIPS1 perhaps due to the large seasonal variation in $PM_{2.5}$ concentrations. Emission changes are more important than climate changes at AREN1, but it is not clear which is more important at SIPS1 or LAVO1.

(3) The more detailed information regarding the representativeness of the three selected stations are included in lines 97-106 as follows:

We chose three representative stations in different parts of the country to investigate the relation between AWA and $PM_{2.5}$ in detail. The IMPROVE station names are AREN1 (Arendtsville, Pennsylvania; 39.92 °N, 77.31°W; in the Northeast), SIPS1 (Sipsey Wilderness, Alabama; 34.34°N, 87.34°W; in the Southeast) and LAVO1 (Lassen Volcanic NP, California; 40.54°N, 121.58°W; in the West), which are shown with the red dots in Figure 1. They match with the site PSU106 (40.72°N, 77.93°W, in the Northeast), SND152 (34.29°N, 85.97°W, in the Southeast) and LAV410 (40.54°N, 121.58°W, in the West) respectively, which has differing impacts of meteorological persistence on the distribution and extremes of ozone in Sun et al. (2017) to allow comparison between the ozone and $PM_{2.5}$ response to AWA. Long range transport from Asia and meteorology are dominant drivers of pollutants at LAVO1, where anthropogenic influence is at a minimum as a clean air site in California (Vancure et al., 2015).

Reference

Sun, W., Hess, P., and Liu, C.: The impact of meteorological persistence on the distribution and extremes of ozone, Geophys. Res. Lett., 44, 1545–1553, 2017.

Parrish, D. D., et al.: Long-term changes in lower tropospheric baseline ozone concentrations at northern mid-latitudes, Atmos. Chem. Phys., 12, 11,485–11, 504, 2012.

VanCuren, R. and Gustin, M. S.: Identification of sources contributing to PM2.5 and ozone at elevated sites in the western U.S. by receptor analysis: Lassen Volcanic National Park, California, and at Great Basin National Park, Nevada, Sci. Total Environ., 530, 505–518, 2015.

Figure 4: What is the meaning of the contour lines in this figure?
Response: Thank you for pointing out this was not explained well: the contour lines are composite 500 hPa geopotential height anomaly (positive values are represented by solid green lines and negative values by dashed magenta lines). They are described in caption for Figure 4 as follows:

Figure 4. (Contour) composite 500 hPa geopotential height anomaly (positive values are represented by solid green lines and negative values by dashed magenta lines) and (shaded) regression coefficients between daily AWA and $PM_{2.5}$ at site (denoted by the black dots) (a, d, g) AREN1, (b, e, h) SIPS1 and (c, f, i) LAVO1 in the study domain for daily JJA time series of current climates. The top row are results using IMPROVE $PM_{2.5}$ and reanalysis AWA, the middle row uses the reanalysis driven simulated $PM_{2.5}$ (REFC1SD) and reanalysis AWA, and the bottom row uses current climate simulated $PM_{2.5}$ and AWA (GCM2000). Stippling indicates the regions that are statistically significant at the 95% confidence level. Unit: $10^{-8}\,\mu\,g\,m^{-3}\,/m^2$ for regression coefficients.

Section 3.1: "averaging together all AWA corresponding to daily PM2.5 above the 90th quantile shows a similar strong correlation between PM2.5 and AWA" Unless I'm missing something, I don't think "correlation" is the correct word here (and elsewhere in this section). It suggested to me that an additional regression was being performed for days with PM2.5 > 90th percentile, but I don't think that is correct. Please clarify -- without additional information on behavior of other PM2.5

percentiles, we can't really define correlation from this subset of daily conditions.

On a related note, I don't really see the benefit of including/discussing both regressions (Fig 4) and high PM2.5 filtered subset (Fig 5). What is gained from this comparison, beyond noting that they are somewhat consistent? It seems neither surprising nor useful to me.

Response:

(1) Thank you for pointing that this could be confused. Here 'correlation', 'correlate' are revised as 'connection' and 'connect', respectively, in discussion for Figure 5 in lines 251-253, 258-259 as follows:

The composite AWA calculated by averaging together all AWA corresponding to daily $PM_{2.5}$ above the $90^{th}$ quantile shows a similar strong **connection** between $PM_{2.5}$ and AWA as that seen for the average (Figure 4 vs. 5).

Overall, the composite AWA for $PM_{2.5}$ also shows that the daily $PM_{2.5}$ above its $90^{th}$ quantile **connects** strongly with AWA during summer.

(2) The composite AWA for $PM_{2.5}$ larger than its quantiles ranging from $30^{th}$ to $90^{th}$ is shown in Figure S2 as follows. The pattern for the composite AWA corresponding to daily $PM_{2.5}$ above the $90^{th}$ quantile is most similar to regression coefficients between daily AWA and $PM_{2.5}$. So we can say that the composite AWA calculated by averaging together all AWA corresponding to daily $PM_{2.5}$ above the $90^{th}$ quantile shows a similar strong connection between $PM_{2.5}$ and AWA as that seen for the average.

We show both approaches to show that they are similar in their spatial distribution, thus suggesting a robust metric. The reason for choosing $90^{th}$ quantile is included in line 249-251 as follows:

The pattern for the composite AWA corresponding to daily $PM_{2.5}$ above the $90^{th}$ quantile is most similar to regression coefficients between daily AWA and $PM_{2.5}$ by comparing different quantiles (Figure S1). The composite AWA calculated by averaging together all AWA corresponding to daily $PM_{2.5}$ above the $90^{th}$ quantile shows a similar strong connection between $PM_{2.5}$ and AWA as that seen for the average (Figure 4 vs. 5).

[Figure]

[Figure]

(c)                                          (d)

[Figure]

Figure S1. (Shaded) composite AWA for PM$_{2.5}$ larger than (a) 30$^{th}$ quantile, (b) 40$^{th}$ quantile, (c) 50$^{th}$ quantile, (d) 60$^{th}$ quantile, (e) 70$^{th}$ quantile, (f) 80$^{th}$ quantile and (g)90$^{th}$ quantile; (h) (shaded) regression coefficients between daily AWA and PM$_{2.5}$ at site AREN1. Unit: $10^{8}$ m$^{2}$ for AWA and $10^{-8}$ μ g m$^{-3}$ /m$^{2}$ for regression coefficients.

Figures 6 and 7b: The diverging colorbar used in these figures does not appear to be appropriate, as the neutral/lighter shades and color split does not occur around zero. The choice creates an artificial split dividing the maps between red and blue that does not seem to have any purpose. Please change to a single color option, or equivalent. Figure 7b also needs more work and cleanup, especially with its legend and units.

Response:

Figure 6. In order to keep consistency in color with Figure 4 and Figure 5, the colorbar for Figure 6 is selected to between red and blue. The colorbar is a little bit sudden if it is in other colors as follows.

[Figure]

Figure 6. The maximum of the composite AWA distribution for PM$_{2.5}$ larger than 90$^{th}$ quantile(Shaded) (a, c, e, g, i), and the centers of the spatial regression coefficient distribution between PM$_{2.5}$ and AWA (b, d, f, h, j): observations (Obs, first row), current climate from the reanalysis driven simulation (REFC1SD, second row), current climate from the coupled model simulation (GCM2000, third row), future climate with current emission (GCM2100, fouth row) and future climate with future emission (REFC2, bottom row). At each grid point, the highest composite AWA anywhere in the domain based on the PM$_{2.5}$ larger than 90$^{th}$ quantile and the highest regression coefficient between AWA and PM$_{2.5}$ are shown. In a) and b), the thee representative sites are denoted by the black dots. In a) and b) the different shapes (circle or triangle) indicate the number of values for every grid that are statistically significant (at the 95% confidence level) is more than 30% or not. The different colors indicate different highest composite AWA and regression coefficients as indicated in the legend. In c) through j) the number of values for every grid that are statistically significant at the 5% confidence level are shown (in black contours).

==mark==Figure 7b was replotted with a smaller difference in legend based on 5% significance level between 90$^{th}$ and 50$^{th}$ percentile.==mark==

[Figure]

==mark==Figure 7. (b) The subtraction of 50th percentile quantile regression slope from 90th percentile quantile regression slope between PM$_{2.5}$ and impact region's average AWA across all 150 sites in the US (at the 5% significance level).==mark==

Section 3.3, quantile regression: The increasing sensitivity for higher percentiles is interesting and potentially important, but it is also important to acknowledge the huge range and overall poor predictive power in this correlation. An r of 0.36 implies that the vast majority (nearly 90%) of all variability is being driven by factors other than AWA at the AREN1 site. Alongside the examination of increasing quantile sensitivities, it must be noted that this lack of overall correlation implies other (and likely MUCH more important) drivers of PM2.5 variability at sites such as this. This is one area in particular where introducing PM2.5 speciation may prove crucial for understanding differences in bulk aerosol behavior.

Response:

The linear regression (r of 0.36) implies that the vast majority (nearly 90%) of all variability is being driven by factors other than AWA at the AREN1 site, which is consistent with the very small R$^2$ at this site in Figure 8.

There are no data on speciation of $PM_{2.5}$ in the model runs in this study. The relationship between $PM_{2.5}$ speciation and AWA using IMPROVE and available model runs will be examined particularly at AREN1 in future to understand differences in bulk aerosol behavior.

The explanation for the r between JJA deseasonalized $PM_{2.5}$ and impact region's average AWA at the AREN1 site is included in line 298-300 as follows:

The correlation coefficient of 0.36 between JJA deseasonalized $PM_{2.5}$ and impact region's average AWA implies that the vast majority of all variability is being driven by factors other than AWA at the AREN1 site. It must be noted that this lack of overall correlation implies other drivers of $PM_{2.5}$ variability at sites like this.

Section 3.3, quantile regression differences: This comparison and examination of Figure 7b seems a little unclear and potentially misleading, as a very high fraction of sites are in the [0 1) bin, suggesting a very small (and potentially insignificant) difference between 50th and 90th percentiles. A better choice of colors is needed to distinguish tiny differences (of either sign). More robust tools to evaluate and compare quantile regression coefficients are available, and should be used to establish significance here.

Response:

A small difference based on the 5% significance level between $90^{th}$ and $50^{th}$ percentile is used in Figure 7b as follows. The numbers for '<=0', '[0 0.5)', '[0.5 1)', '[1 1.5)', '[1.5 2)', '[2 2.5)', '>=2.5' are 5, 18, 34, 25, 26, 18 and 24, respectively.

A Z-test is used to compare two quantile regression slopes for $90^{th}$ and $50^{th}$ percentile at the 5% significance level between $PM_{2.5}$ and impact region's average AWA across all 150 sites in the US. The formula for this statistical test (see Paternoster et al., 1998 and Clogg et al., 1995 for more discussion) is:

$$Z = \frac{b_1 - b_2}{\sqrt{(SE_{b1}^2 + SE_{b2}^2)}} \qquad (1)$$

Where $b_1$ and $b_2$ are quantile regression slopes for 90th and $50^{th}$ percentile, $SE_{b1}$ and $SE_{b2}$ are standard errors for $b_1$ and $b_2$. Calculate the value of Z for every grid using equation (1). The difference between $90^{th}$ percentile quantile regression slope and $50^{th}$ percentile quantile regression slope is significant at the 5% significance level if Z is great than +/-1.96 across all 150 sites in the US.

[Figure]

Figure 7. (b) The subtraction of 50th percentile quantile regression slope from 90th percentile quantile regression slope between PM$_{2.5}$ and impact region's average AWA across all 150 sites in the US (at the 5% significance level).

References

Clogg, C. C., Eva, P., and Adamantios, H.: Statistical methods for comparing regression coefficients between models, American Journal of Sociology, 100:1261-1293, 1995.

Paternoster, R., Brame, R., Mazerolle, P., and Piquero, A.: Using the correct statistical test for the equality of regression coefficients, Criminology, 36(4), 859-866. https://doi.org/10.1111/j.1745-9125.1998.tb01268.x., 1998.

Section 3.4: This section all became very hand-wavey to me. I'm not clear on many of the decisions, methods, and conclusions being made here, and more explanation and discussion would be appropriate. In particular, I don't understand the switch to "interannual variance", nor how that was calculated. Previous examinations looked at daily variability -- why the switch to interannual? How many years were included in this calculation? 75% explained variability seems VERY high to my eye, given previous figures and results. I would need to see more time devoted to explaining how this regression was put together before I could accept it at face value. The spatial inconsistency with other studies is also of concern. How robust can we assume this result to be, given the apparent inconsistencies? What does it mean to say "where meteorology dominates", and how is this defined?

Response:

(1) "interannual variance":

We calculated daily and interannual variability, respectively. Applying daily change in PM$_{2.5}$ and AWA, the coefficient of determination ($R^2$) is up to 0.56. While the coefficient of determination ($R^2$) is up to 0.75 using interannual variability in PM$_{2.5}$ and AWA. So here we focused on interannual variability in PM$_{2.5}$ and AWA.

(2) "The years and calculation process":

The coefficient of determination ($R^2$) is calculated from the linear regression model (equation (6) : $PM_{2.5} = \beta \cdot p + \alpha$) using simulated PM$_{2.5}$ and AWA for the present climate (GCM2000, 2006-2025, 20 years). The change of PM$_{2.5}$ (denoted by ΔPM$_{2.5}$) in the future (GCM2100, 2106-2125, 20 years) due to the change in AWA (GCM2100-GCM2000) is calculated using the equation ((7): $\Delta PM_{2.5} = \beta[(\overline{AWA}_f - \overline{AWA}_p) \cdot S]$ ), where β and S are calculated from the values for the present climate (GCM2000, 2006-2025, 20 years).

(3) "where meteorology dominates":

Figure 2 shows that emission changes are not more important than climate changes at LAVO1 (in western US). So here the western US is the area where meteorology dominates.

(4) In order to make this section more clear, the period is included in line 315 as follows:

The strong association between PM$_{2.5}$ concentrations and AWA in the current climate prompts us to investigate the extent to which we can utilize a linear regression model to predict changes in PM$_{2.5}$ concentrations from AWA change in future climate. Employing daily present-day summertime concentrations of PM$_{2.5}$ and AWA for current climate from the coupled model simulation (GCM2000, 2006-2025) and equation (5)-(6), we derive that how much of PM$_{2.5}$'s interannual variance can be explained by the projection of JJA AWA anomalies onto the daily PM$_{2.5}$-

AWA regression coefficients pattern. The coefficient of determination ($R^2$) of the linear regression model using simulated $PM_{2.5}$ and AWA for the present climate varies from 0 to 0.75 depending on gridbox (Figure 8). This means that the projected value (using only AWA changes) captures up to 75% of the interannual variability in $PM_{2.5}$ over Great Plains and West. Wise and Comrie (2005) similarly determined $R^2$ values of 0.1-0.5 for associations of PM with atmospheric variables across sites in the Southwest. Because of the high correlation coefficients (75%) this suggests that the regression results reveal the broad population instead of a small number of influential outliers (Cook, 1979). The $R^2$ measures the part of variance of $PM_{2.5}$ that can be explained by the linear regression model (Kutner et al., 2004).

Citation: https://doi.org/10.5194/acp-2021-750-RC2

---

## Author Response (AR2)

The authors have put in considerable effort for their revisions, and for the most part I am satisfied with their changes. The one point on which I still feel additional work is necessary is on the issue of PM2.5 speciation.

I understand the problem of data availability, and in my original review I was not suggesting that the authors create robust longterm speciated measurements where none exist. What I do (still) expect to see, however, is some significant discussion of this issue and how spatiotemporal patterns of PM2.5 speciation may affect the analyses and projections performed here. While we do lack much of the data necessary for a fully robust speciated analysis as performed in this manuscript, we are not completely blind with respect to how different types of PM2.5 precursors vary regionally, and much work has already explored differences in their expected behavior under varying ambient conditions. As just one example, Tai et al., 2010 (already cited in the manuscript) explore ways in which PM2.5 from sulfate and nitrate differ both in spatial distributions as well as their respective response to meteorology. Considering the significance of these differences, and the availabilitiy of previous studies like this one that have addressed them, I find it hard to accept the lack of meaningful discussion on the topic in this manuscript.

Furthermore, modern climate and chemical transport models (including CESM and the CAM4 atmospheric component) provide this speciation as gridded output, further supporting analysis on the model side. While full validation against observations may be outside the scope of this work, at the very least this output could aid in the interpretation of the model results themselves. Why has this not been done with the model output used here?

With this point addressed, I would feel comfortable giving full support for publication in ACP.

Response:
    We would like to thank the reviewer for the constructive comments about the manuscript and we agree, and add the following text to the paper in lines 379-390 (at the end of Discussion section):
    $PM_{2.5}$ generally consists of multiple different aerosols each with different sources and variability; for example, the most important in the US are sulfate, organic matter, elemental carbon, nitrate, ammonium and desert dust. The different $PM_{2.5}$ components respond to meteorological variables differently. The sulfate fraction of $PM_{2.5}$ is predicted to be higher due to faster $SO_2$ oxidation under a warmer climate while the nitrate and organic fraction lower due to volatility (Dawson et al., 2007; Kleeman, 2008; Tai et al., 2010). Increased temperatures can lead up to higher biogenic emissions of $PM_{2.5}$ precursors including agricultural ammonia, soil $NO_x$, and volatile organic compounds (Pinder et al., 2004; Bertram et al., 2005; Guenther et al., 2006; Riddick et al., 2016). Aqueous-phase sulfate and ammonium nitrate production increase with higher relative humidity (Liao et al., 2006; Dawson et al., 2007). Wildfires are an important source of black and organic carbon and they can increase or decrease depending on the local changes in climate and land use (Park et al., 2007; Spracklen et al., 2009; Kloster et al., 2012). Future exploration of the different components of aerosols and how each

responds to climate could provide more information about the effect on each type, but for these simulations, only PM$_{2.5}$ was output and thus is not available for this study.

References:

Bertram, T. H., Heckel, A., Richter, A., Burrows, J. P., and Cohen, R. C.: Satellite measurements of daily variations in soil NOx emissions, Geophys. Res. Lett., 32, L24812, https://doi.org/10.1029/2005gl024640, 2005.

Dawson, J. P., Adams, P. J., and Pandis, S. N.: Sensitivity of PM$_{2.5}$ to climate in the Eastern US: a modeling case study, Atmos. Chem. Phys., 7, 4295–4309, https://doi.org/10.5194/acp-7-4295-2007, 2007.

Guenther, A., Karl, T., Harley, P., Wiedinmyer, C., Palmer, P. I., and Geron, C.: Estimates of global terrestrial isoprene emissions using MEGAN (Model of Emissions of Gases and Aerosols from Nature), Atmos. Chem. Phys., 6, 3181–3210, https://doi.org/10.5194/acp-6-3181-2006, 2006.

Kleeman, M. J.: A preliminary assessment of the sensitivity of air quality in California to global change, Clim. Change, 87, S273–S292, https://doi.org/10.1007/S10584-007-9351-3, 2008.

Kloster, S., Mahowald, N. M., Randerson, J. T., and Lawrence P. J.: The impacts of climate, land use, and demography on fires during the 21st century simulated by CLM-CN, Biogeosciences, 9(1), 509–525, 2012.

Liao, H., Chen, W. T., and Seinfeld, J. H.: Role of climate change in global predictions of future tropospheric ozone and aerosols, J. Geophys. Res.-Atmos., 111, D12304, https://doi.org/10.1029/2005jd006852, 2006.

Park, R. J., Jacob, D. J., and Logan, J. A.: Fire and biofuel contributions to annual mean aerosol mass concentrations in the United States, Atmos. Environ., 41, 7389–7400, 2007.

Pinder, R. W., Pekney, N. J., Davidson, C. I., and Adams, P. J.: A process-based model of ammonia emissions from dairy cows: Improved temporal and spatial resolution, Atmos. Environ., 38, 1357–1365, 2004.

Riddick, S., Ward, D., Hess P., Mahowald, N., Massad R., and Holland E.: Estimate of changes in agricultural terrestrial nitrogen pathways and ammonia emissions from 1850 to present in the Community Earth System Model, Biogeosciences, 13(11), 3397–3426, 2016.

Spracklen, D. V., Mickley, L. J., Logan, J. A., Hudman, R. C., Yevich, R., Flannigan, M. D., and Westerling, A. L.: Impacts of climate change from 2000 to 2050 on wildfire activity and carbonaceous aerosol concentrations in the western United States, J. Geophys. Res.-Atmos., 114, D20301, https://doi.org/10.1029/2008jd010966, 2009.

Tai, A. P. K., Mickley, L. J., and Jacob, D. J.: Correlations between fine particulate matter (PM$_{2.5}$) and meteorological variables in the United States: Implications for the sensitivity of PM$_{2.5}$ to climate change, Atmos. Environ., 44, 3976–3984, 2010.